# Unsupervised changes in core object recognition behavior are predicted by neural plasticity in inferior temporal cortex

Xiaoxuan Jia[1,2†*], Ha Hong[1,2,3‡], James J DiCarlo[1,2,4]

[1]Department of Brain and Cognitive Sciences, Massachusetts Institute of Technology, Cambridge, United States; [2]McGovern Institute for Brain Research, Cambridge, United States; [3]Harvard-MIT Division of Health Sciences and Technology, Cambridge, United States; [4]Center for Brains, Minds and Machines, Cambridge, United States

**Abstract** Temporal continuity of object identity is a feature of natural visual input and is potentially exploited – in an unsupervised manner – by the ventral visual stream to build the neural representation in inferior temporal (IT) cortex. Here, we investigated whether plasticity of individual IT neurons underlies human core object recognition behavioral changes induced with unsupervised visual experience. We built a single-neuron plasticity model combined with a previously established IT population-to-recognition-behavior-linking model to predict human learning effects. We found that our model, after constrained by neurophysiological data, largely predicted the mean direction, magnitude, and time course of human performance changes. We also found a previously unreported dependency of the observed human performance change on the initial task difficulty. This result adds support to the hypothesis that tolerant core object recognition in human and non-human primates is instructed – at least in part – by naturally occurring unsupervised temporal contiguity experience.

**\*For correspondence:**
jxiaoxuan@gmail.com

**Present address:** †Allen Institute for Brain Science, Seattle, United States; ‡Caption Health, 2000 Sierra Point Pkwy, Brisbane, United States

**Competing interests:** The authors declare that no competing interests exist.

## Introduction

Among visual areas, the inferior temporal (IT) cortex is thought to most directly underlie core visual object recognition in human and non-human primates (*Ito et al., 1995*; *Rajalingham and DiCarlo, 2019*). For example, simple weighted sums of IT neuronal population activity can accurately explain and predict human and monkey core object recognition (COR) performance over dozens of such tasks (*Majaj et al., 2015*). Moreover, direct suppression of IT activity disrupts COR behavior (*Afraz et al., 2015*; *Rajalingham and DiCarlo, 2019*). These results were found in the face of significant variation in object latent variables including size, position, and pose, and the high performance of the simple IT readout (weighted sum) rests on the fact that many individual IT neurons show high tolerance to those variables (*DiCarlo et al., 2012*; *Hung et al., 2005*; *Li et al., 2009*), reviewed by *DiCarlo et al., 2012*.

But how does the ventral stream wire itself up to construct these highly tolerant IT neurons? Simulated IT 'neurons' in the deep layers of artificial neural networks (ANNs) have such tolerance and provide quite accurate approximations of the adult ventral visual stream processing (*Khaligh-Razavi and Kriegeskorte, 2014*; *Rajalingham et al., 2018*; *Yamins et al., 2014*). However, those ANNs are produced by training with millions of supervised (labeled) training images, an experience regime that is almost surely not biologically plausible over evolution or postnatal development. That simple fact rejects all such ANNs as models of the construction of IT tolerance, regardless of whether

**eLife digest** A bear is a bear, regardless of how far away it is, or the angle at which we view it. And indeed, the ability to recognize objects in different contexts is an important part of our sense of vision. A brain region called the inferior temporal (IT for short) cortex plays a critical role in this feat. In primates, the activity of groups of IT cortical nerve cells correlates with recognition of different objects – and conversely, suppressing IT cortical activity impairs object recognition behavior. Because these cells remain selective to an item despite changes of size, position or orientation, the IT cortex is thought to underly the ability to recognise an object regardless of variations in its visual properties. How does this tolerance arise?

A property called 'temporal continuity' is thought to be involved – in other words, the fact that objects do not blink in and out of existence. Studies in nonhuman primates have shown that temporal continuity can indeed reshape the activity of nerve cells in the IT cortex, while behavioural experiments with humans suggest that it affects the ability to recognize objects. However, these two sets of studies used different visual tasks, so it is still unknown if the cellular processes observed in monkey IT actually underpin the behavioural effects shown in humans. Jia et al. therefore set out to examine the link between the two.

In the initial experiments, human volunteers were given, in an unsupervised manner, a set of visual tasks designed similarly to the previous tests in nonhuman primates. The participants were presented with continuous views of the same or different objects at various sizes, and then given tests of object recognition. These manipulations resulted in volunteers showing altered size tolerance over time. Aiming to test which cellular mechanism underpinned this behavioural effect, Jia et al. built a model that simulated the plasticity of individual IT cells and the IT networks, to predict the changes of object recognition observed in the volunteers. A high predictability of the model revealed that the plasticity in IT cortex did indeed account for the behavioral changes in the volunteers. These results shed new light on the role that temporal continuity plays in vision, refining our understanding of the way the IT cortex helps to assess the world around us.

or not the brain is executing some form of backpropagation-instructed plasticity (*Lillicrap et al., 2020*; *Rumelhart et al., 1986*). So the question remains open: how does the ventral stream wire itself up to construct a tolerant IT with minimal supervision?

The temporal stability of object identity under natural viewing (i.e., objects do not rapidly jump in and out of existence) has been proposed as a key available source of unsupervised information that might be leveraged by the visual system to construct neural tolerance, even during adulthood (*Földiák, 1991*; *Hénaff et al., 2019*; *Rolls and Stringer, 2006*; *Wallis et al., 2009*; *Wallis et al., 2009*; *Wiskott and Sejnowski, 2002*). Consistent with this view, psychophysical results from human subjects show that unsupervised exposure to unnatural temporal contiguity experience (i.e., laboratory situations in which object *do* jump in and out of existence) reshapes position tolerance (*Cox et al., 2005*), pose tolerance (*Wallis and Bülthoff, 2001*), and depth illumination tolerance (*Wallis et al., 2009*) as measured at the behavioral level. Similarly, neurophysiological data from adult macaque IT show that unsupervised exposure to unnatural temporal contiguity experience reshapes IT neuronal position and size tolerance (*Li et al., 2009*; *Li and DiCarlo, 2008*; *Li and DiCarlo, 2010*), in a manner that is qualitatively consistent with the human behavioral data.

Taken together, our *working hypothesis* is that the ventral visual stream is under continual reshaping pressure via unsupervised visual experience, that such experience is an important part of the construction of the tolerant representation that is ultimately exhibited at the top level of the ventral stream (IT), that the IT population feeds downstream causal mechanistic chains to drive core object discrimination behavior, and that the performance on each such behavioral tasks is well approximated by linear readout of IT (*Hung et al., 2005*; *Majaj et al., 2015*).

However, there is a key untested prediction in this working hypothesis: is the single neuronal plasticity in adult monkey IT quantitatively consistent with the adult human behavioral changes resulting from unsupervised temporal contiguity experience? In this study, we chose to focus on testing that missing link as it was far from obvious that it would hold up. In particular, the prior IT neurophysiology work was with basic-level objects and produced seemingly large changes (~25% change in IT

selectivity per hour of exposure in *Li and DiCarlo, 2010*), and the prior human behavioral work was with subordinate-level objects and produced significant, but subtle, changes in behavior (e.g., ~3% performance change in *Cox et al., 2005*). Moreover, if we found that the link did not hold, it would call into question all of the elements of the overall working hypothesis (especially IT's relationship to COR behavior,and/or the importance of unsupervised plasticity to the IT representation). Thus, either result would be important.

To test whether our *working hypothesis* is quantitatively accurate over the domain of unsupervised temporal contiguity-induced plasticity, we sought to build a model to predict the changes in human object discrimination performance that should result from temporally contiguity experience-driven changes in IT neuronal responses. This model has three components: (1) a generative IT model (constrained by prior IT population response; *Majaj et al., 2015*) that approximates the IT population representation space and can thus simulate the IT population response to any image of the objects (within the space) with variation in size; (2) an unsupervised plasticity rule (constrained by prior IT neural plasticity data; *Li and DiCarlo, 2010*) to quantitatively describe and predict firing rate (FR) change of single IT neurons resulting from temporally contiguous pair of experienced images and can thus be used to update the simulated IT population representation; and (3) an IT-to-COR-behavior-linking model (learned weighted sums, previously established by *Majaj et al., 2015*) to predict behavioral discrimination performance from the state of the IT (simulated) population both before and after each epoch of unsupervised experience.

To overcome the limitation of non-overlapping tasks in previous psychophysics and neurophysiology studies and to extend prior psychophysical work, we carried out new human behavioral experiments. Specifically, we measured the progression of changes in size-specific human object discrimination performance that resulted from unsupervised temporal contiguity experience using the same exposure paradigm as the prior monkey neurophysiology work (*Li and DiCarlo, 2010*). We did not use the exact same images as prior work, but we expected the model to still make accurate predictions of all behavioral changes. We made these behavioral measurements for a wide range of object discrimination tasks, ranging from subordinate (specifically different face objects) to basic level.

Because humans make sensory-independent mistakes due to inattentional state, these sensory-independent random choices (referred to as lapse rate) set a ceiling in the measurable human behavioral performance (*Prins, 2012*; *Wichmann and Hill, 2001*). When tasks are in the saturated regime, it is hard to detect any learning effect as any changes in sensory representation would be hidden by the behavioral ceiling (see later). Therefore, we focused our psychophysical study in the mid-range of task difficulty where learning effects can be measured. However, this meant that the task difficulty in human psychophysics could not be in the basic object regime where the neural data were collected. Thus, to make behavioral predictions from the neural data, we took advantage of the overall model to build this bridge: we first tuned the unsupervised plasticity rule by neural data with basic-level object images (*Li and DiCarlo, 2010*); we then used a generative IT model – capable of simulating the response of each artificial IT neuron for a wide range of image discriminability levels – to make quantitative predictions of behavioral change in the regime where the human behavioral learning effects can be readily measured.

Indeed, our behavioral tests revealed a strong dependency of learning effect on the initial task difficulty, with initially hard (d' < 0.5) and initially easy (d' > 2.5) COR tasks showing smaller measured learning effects than COR tasks of intermediate initial difficulty. We found that our overall model was quite accurate in its predictions of the direction, magnitude, and time course of the changes in measured human size tolerance in the regime where behavioral effects were readily measured for all of the tested unsupervised experience manipulations. The overall model also predicted how the behavioral effect size depended on the initial d' once we assume behavioral lapses (*Prins, 2012*) in the model at approximately the same level as those inferred in our subject pool. We note that, because of the (expected) inability to observed behavioral changes for tasks with initial high d', this study cannot confirm or refute the hypothesized linkage between IT neural effects and behavioral effects in that particular regime.

Taken together, this result shows that at least three separate types of studies (human unsupervised learning, IT unsupervised plasticity, and IT-to-COR-behavior testing) are all quantitatively consistent with each other. As such, this result adds support to the overall working hypothesis: that tolerant COR is instructed – at least in part – by naturally occurring unsupervised temporal contiguity

experience that gradually reshapes the non-linear image processing of the ventral visual stream without the need for millions of explicit supervisory labels (*Krizhevsky et al., 2017*; *LeCun et al., 1989*; *Riesenhuber and Poggio, 1999*) and reviewed by *LeCun et al., 2015*.

## Results

Our working hypothesis (see Introduction) predicts that IT population plasticity resulting from unsupervised visual experience should accurately predict the direction, magnitude, and time course of all changes in human object discrimination performance resulting from the same visual exposure. To quantitatively test these predictions, we first carried out a set of human psychophysical experiments with unsupervised temporal continuity experience that closely approximate the exposure paradigm that has been shown to reliably produce IT plasticity (measured as changes in size tolerance at single IT recording site) (*Li and DiCarlo, 2010*).

### Measure changes in human object discrimination performance induced by unsupervised visual experience

The basic experimental strategy is that, after testing initial object discrimination performance on a set of discrimination tasks ('*Test phase*,' *Figure 1A*), we provide an epoch of unsupervised visual experience ('*Exposure phase*,' *Figure 1A*) that is expected to result in IT plasticity (based on the results of *Li and DiCarlo, 2010*). At the end of the exposure epoch, we remeasure discrimination performance (*Test phase*), then provide the next epoch of unsupervised experience (*Exposure phase*), etc. (see *Figure 1A*). This strategy allowed us to evaluate the accumulation of positive or negative behavioral changes (a.k.a. 'learning') resulting from four unsupervised experience epochs (400 exposure 'trials' each) over approximately 1.5–2 hr. We include control discrimination tasks to subtract out any general learning effects.

Specifically, we evaluated changes in discrimination performance (relative to initial performance) of each of a set of size-specific object discrimination tasks. A total of 174 human subjects on Amazon Mechanical Turk (see Materials and methods and *Kar et al., 2019*; *Majaj et al., 2015*; *Rajalingham et al., 2018*) participated in this experiment.

To measure object discrimination performance in each subject, we used a set of two-way alternative forced choice (2AFC) sub-tasks (size-specific object discrimination tasks; see Materials and methods). These sub-tasks were randomly interleaved (trial by trial) in each test phase, and the key test conditions used in the analyses (brackets indicated with d's in *Figure 1B*) were embedded within a balanced set of six sub-tasks and cover trials (see *Figure 1B* and Materials and methods).

Our first experiments used pairs of faces as the objects to discriminate, and we targeted our exposure manipulations at the big size (2× the baseline size; see Materials and methods and *Figure 1*; later, we targeted other pairs of objects and other sizes). Specifically, we used eight face objects from a previous study (*Majaj et al., 2015*). We chose these face objects at this size because, prior to unsupervised exposure, they had intermediate discriminability (mean d' = 2.0 ± 0.1 for big size, frontal view, n = 28 pairs of faces), thus allowing us the possibility to measure both positive and negative changes in discrimination performance. For each subject, two target faces (manipulated during exposure) and two control faces (not shown during exposure) were randomly chosen from these eight faces.

Subjects were instructed to identify the single foreground face in a briefly presented test image (100 ms) by choosing among two alternative choice faces immediately presented after the test image, one of which is always correct (i.e., 50% chance rate). The test image contained one foreground object with variation in view (position, size, pose), overlaid on a random background (see Materials and methods for test image generation). The choice images were always baseline views (i. e., size of ~2°, canonical pose) without background.

Similar to prior work testing the effects of unsupervised exposure on single-site IT recordings (*Li and DiCarlo, 2010*), each experiment consisted of two phases (*Figure 1A*): test phases to intermittently measure the size-specific object discrimination performance (d') for the target face pair and control face pair (three d' measured in each group of subjects, see *Figure 1B* bottom); and exposure phases to provide unsupervised visual experience (pairs of images with different sizes in

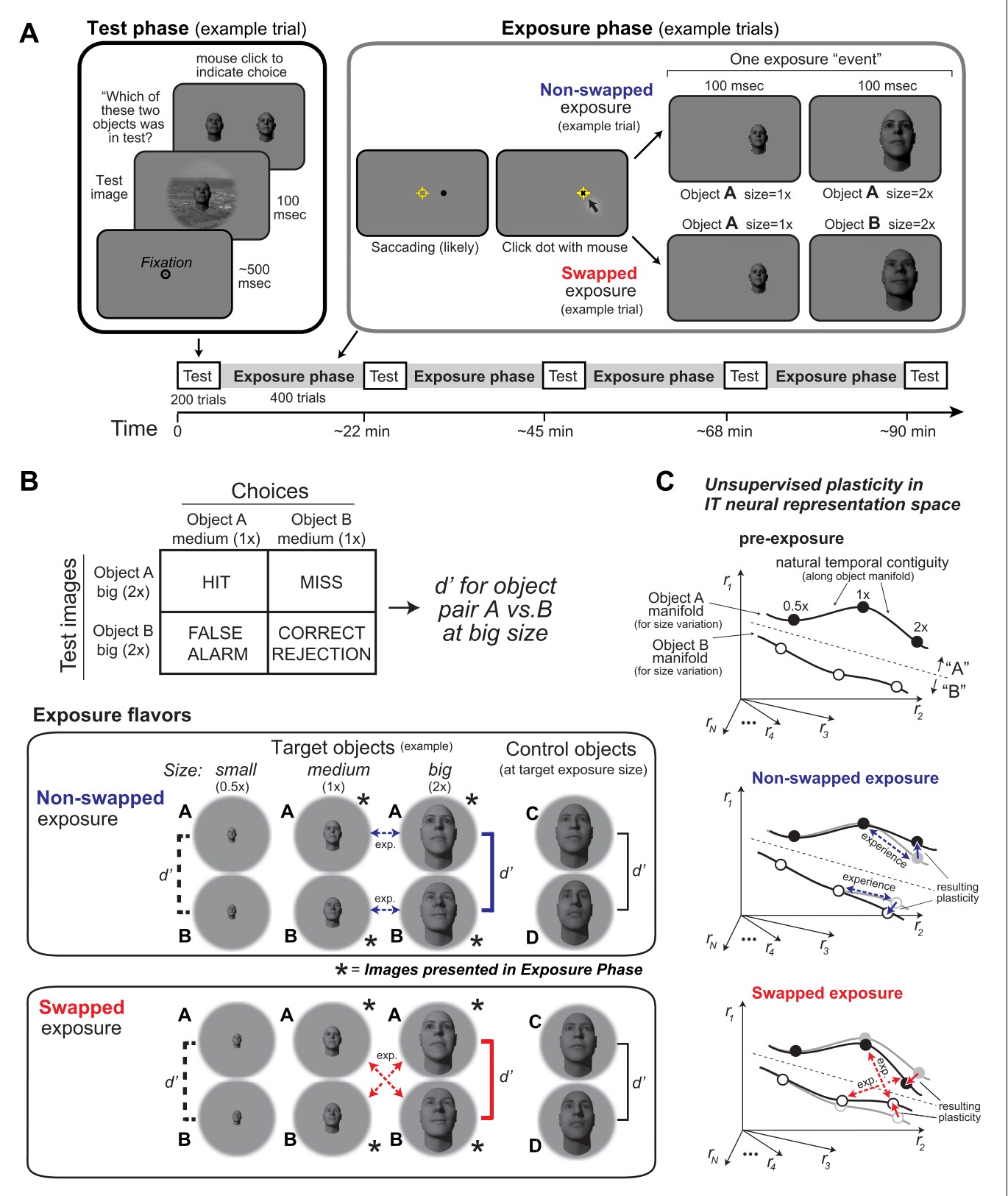

**Figure 1.** Experimental design and conceptual hypothesis. (**A**) Illustration of human behavioral experimental design and an example trial from the Test phase and from the Exposure phase. (**B**) Top: example confusion matrix for a two-way alternative forced choice (2AFC) size-specific sub-task run during each Test phase to monitor object-specific, size-specific changes in discrimination performance (see Materials and methods). Bottom: the two

*Figure 1 continued on next page*

*Figure 1 continued*

unsupervised exposure flavors deployed in this study (see Materials and methods). Only one of these was deployed during each Exposure phase (see *Figure 2*). Exposed images of example-exposed objects (here, faces) are labeled with asterisks, and the arrows indicate the exposure events (each is a sequential pair of images). Note that other object and sizes are tested during the Test phases, but not exposed during the Exposure phase (see d' brackets vs. asterisks). Each bracket with a d' symbol indicates a preplanned discrimination sub-task that was embedded in the Test phase and contributed to the results (*Figure 2*). In particular, performance for target objects at non-exposed size (d' labeled with dashed lines), target objects at exposed size (d' labeled with bold solid lines), and control objects (d' labeled with black line) was calculated based on test phase choices. (C) Expected qualitative changes in the inferior temporal (IT) neural population representations of the two objects that results from each flavor of exposure (based on *Li and DiCarlo, 2010*). In each panel, the six dots show three standard sizes of two objects along the size variation manifold of each object. Assuming simple readout of IT to support object discrimination (e.g., linear discriminant, see *Majaj et al., 2015*), non-swapped exposure tends to build size-tolerant behavior by straightening out the underlying IT object manifolds, while swapped exposure tends to disrupt ('break') size-tolerant behavior by bending the IT object manifolds toward each other at the swapped size. This study asks if that idea is quantitatively consistent across neural and behavioral data with biological data-constrained models.

The online version of this article includes the following figure supplement(s) for figure 1:

**Figure supplement 1.** Outline and example test images.

close temporal proximity; *Figure 1A*) that – based on prior work – was expected to improve or decrease the discrimination performance on the exposed objects.

The purpose of the exposure phase was to deploy unsupervised visual experience manipulations to target a particular object pair (two 'target' objects) at particular views (e.g., sizes) of those target objects. For each exposure event, two images, each containing a different size object (frontal; no background), were presented consecutively (100 ms each) (see Materials and methods for details). In non-swapped exposure events, both images contained the same object (expected to 'build' size tolerance under the temporal contiguity hypothesis). In swapped exposure events, each images contained a different target object (expected to 'break' size tolerance under the temporal contiguity hypothesis). The conceptual predictions of the underlying IT neural population target object manifolds (*DiCarlo and Cox, 2007*) are that non-swapped exposure events will straighten the manifold of each target object by associating size exemplars of the same object (as in the natural world), and that swapped exposure events will bend and decrease the separation between the two manifolds by incorrectly associating size exemplars of different objects (*Figure 1C*). This logic and experimental setup are adopted entirely from prior work (*Li and DiCarlo, 2008*; *Li and DiCarlo, 2010*).

In our studies here, we specifically focused on manipulating the size tolerance in the medium size ($\times 1$ of baseline view; $\sim 2°$) to big size ($\times 2$ of baseline view; $\sim 4°$) regime. Thus, the images shown during the exposure phase (indicated by * in *Figure 1B*) were always medium- and big-size, frontal view of the target objects. We conducted three types of unsupervised exposure experiments (u): swapped (u1), non-swapped (u2) and non-swapped, followed by swapped (u3).

In experiment u1 (swapped exposure events), we found that discrimination of the target face pair viewed at big size decreased with increasing numbers of exposure events (*Figure 2A*; top rows; red solid line; n = 102 subjects). We found little to no change in the performance for the non-exposed (small size) versions of those same faces (black dashed line; mean initial d' is $1.2 \pm 0.1$) or for non-exposed control faces (also tested at big size, black solid line). Lower panels in *Figure 2A* show the learning effect defined by subtracting changes in control face discrimination performance (to remove general learning effects over the experience epochs, which turned out to be small; see *Figure 2A*, upper panel). In sum, we demonstrated an unsupervised, object-selective, size-selective temporal contiguity-induced learning effect that was qualitatively consistent with prior work in 'breaking' tolerance (*Cox et al., 2005*; *Wallis and Bülthoff, 2001*) and measured the accumulation of that learning over increasing amounts of unsupervised exposure.

In experiment u2 (non-swapped exposure events), we found that discrimination of the target face pair viewed at big size *increased* with increasing numbers of exposure events (*Figure 2B*; top rows; blue solid line; n = 36 subjects). As in experiment u1, we found little to no change in performance for the non-exposed (small size) versions of those same faces or for non-exposed control faces (also tested at big size, black solid line). This shows that, as predicted by the temporal contiguity hypothesis, unsupervised experience can build size tolerance at the behavioral level.

Interestingly, after $\sim 800$ exposure events, the exposure-induced learning effects appeared to plateau in both 'breaking' tolerance conditions (experiment u1, *Figure 2A*) and 'building' tolerance

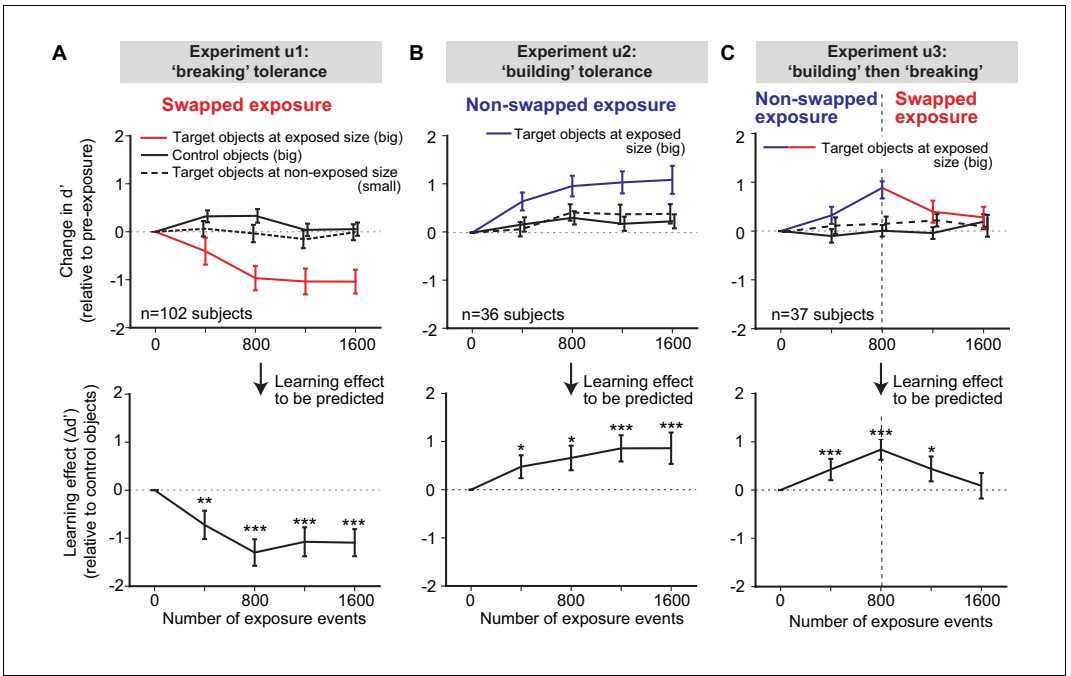

**Figure 2.** Measured human unsupervised learning effects as a function of amount of unsupervised exposure. Each 'exposure event' is the presentation of two, temporally adjacent images (see *Figure 1A*, right). We conducted three longitudinal unsupervised exposure experiments (referred to as u1, u2, and u3). (**A**) Swapped exposure experiment intended to 'break' size tolerance (n = 102 subjects; u1). Upper panels are the changes in d' relative to initial d' for targeted objects (faces) at exposed size (big) (red line), control objects (other faces) at the same size (big) (black line), and targeted faces at non-exposed size (small) (dashed black line) as a function of number of exposure events prior to testing. Lower panel is the to-be-predicted learning effect determined by subtracting change of d' for control objects from the change of d' for target objects (i.e., red line minus black line). (**B**) Same as (**A**), but for non-swapped exposure experiment (n = 36 subjects; u2). (**C**) Same as (**A**), except for non-swapped exposure followed by swapped exposure (n = 37 subjects; u3) to test the reversibility of the learning. In all panels, performance is based on population pooled d' (see Materials and methods). Error bars indicate bootstrapped standard error of the mean population pooled d' (bootstrapping is performed by sampling with replacement across all trials). p-value is directly estimated from the bootstrapped distributions of performance change by comparing to no change condition. * indicates p-value<0.05; ** indicates p-value<0.01; *** indicates p-value<0.001.

conditions (experiment u2, *Figure 2B*), suggesting a limit in the measurable behavioral effects (see Discussion).

To test whether this unsupervised learning effect is reversible, we measured human performance in a combined design (experiment u3) by first providing exposure epochs that should 'build' tolerance, followed by exposure epochs that should 'break' tolerance (n = 37 subjects). Consistent with the results of experiments u1 and u2, we found that size tolerance first increased with non-swapped ('build') exposures and then decreased with swapped ('break') exposures (*Figure 2C*), and that the effect did not spill over to the control objects.

In sum, these results confirmed that the effect of unsupervised visual experience was specific (to manipulated object and sizes) and strong even in adults. Furthermore, the measured human learning effect trajectories with different unsupervised visual exposure conditions (u1, u2, u3) were taken as behavioral effects that must – without any parameter tuning – be quantitatively predicted by our working hypothesis (that links IT neural responses to COR behavior; see Introduction). We next describe how we built an overall computational model to formally instantiate that working hypothesis to make those predictions.

## A generative model to simulate the population distribution of IT responses

To generate predictions of human behavior performance, we need to measure or otherwise estimate individual IT neural responses to the same images used in the human psychophysical testing (above) for a sufficiently large set of IT neurons (a.k.a. IT population responses). Because each of the objects we used in human psychophysics had been previously tested in neural recording experiments from monkey IT, we did not collect new IT population responses (very time consuming), but we decided instead to make suitably accurate predictions of the initial population pattern of IT response for test images of those objects (i.e., the IT response patterns prior to any simulated unsupervised plasticity effects). To do this, we built a generative model of the IT population based on the previously recorded IT population response to those objects. The output of this model is the FR of a simulated IT population to one presentation of a newly rendered test image (generated from the 64 base objects used in the previous study). With this model, we could simulate the initial IT population responses to any image rendered from the psychophysically tested objects (approximately) without recording more neurons in behaving animals.

This generative IT model captures the IT neuronal representation space with a multi-dimensional Gaussian (MDG) model, assuming the distribution of IT population responses is Gaussian-like for each object (see Materials and methods for Gaussian validation) (*Figure 3A*). Because the MDG preserves the covariance matrix of IT responses to 64 objects, any random draw from this MDG gives rise to an object response preference profile (one response level for each of 64 objects) of a simulated IT neural site. To simulate the variance in object size, for each simulated site, we randomly chose one size-tuning kernel from a pool of size-tuning curves that we had obtained by fitting curves to real IT responses across changes in presented object size (n = 168 recording sites; data from *Majaj et al., 2015*). This process is repeated independently for each simulated site. Motivated by prior work (*Li et al., 2009*), we assumed separability of object representation and size tuning, and simulated the response to any of the 64 objects.

To check if the simulation is statistically accurate in terms of the layout of images in IT population representation space, we compared the representation similarity matrix (RSM; correlation between neuronal population responses to different images) of different draws of a simulated IT with the RSM measured from the actual IT neural data (*Figure 3B*). One typical example of that is shown in *Figure 3C*, revealing high correlation of the two RSMs (r = 0.93 ± 0.01). While this does not guarantee that any such simulated IT population is fully identical to an IT population that might exist in an actual monkey or human IT, our goal was simply to get the simulated IT population response distribution in the proper range (up to second-order statistics).

## A standard IT-to-COR-behavior-linking model for core object discrimination behavior

To make predictions about how IT neural changes will result in behavioral changes, we first needed a model to establish the linkage between IT population response and core object discrimination behavior prior to any experience-induced effects. We have previously found that simple weighted linear sums of IT neural responses accurately predict the performance (d') of human object discrimination for new images of those same objects (here termed the IT-to-COR-behavior-linking model) (*Majaj et al., 2015*). That model has only two free hyperparameters: the number of neural sites and the number of labeled (a.k.a. 'training') images used to set the weights of the decoder for each object discrimination. Once those two hyperparameters are locked, it has been empirically demonstrated that the performance for any object discrimination task on new images is accurately predicted by its trained decoder (*Majaj et al., 2015*). To test whether the simulated IT population activity from the generative IT model (above) could quantitatively reproduce those prior results and to lock these two hyperparameters, we compared the predicted performance (for any given object recognition task) based on the simulated IT population (*Figure 3D*; red solid line) with the predicted performance based on the previously recorded IT neural population (black solid line). We did this as a function of number of recording sites for a set of object recognition tasks. *Figure 3D* illustrates two example tasks (error bar is standard error across 40 random subsamples of recording sites). As expected, we found that the model predictions overlapped with decoded performance of real IT

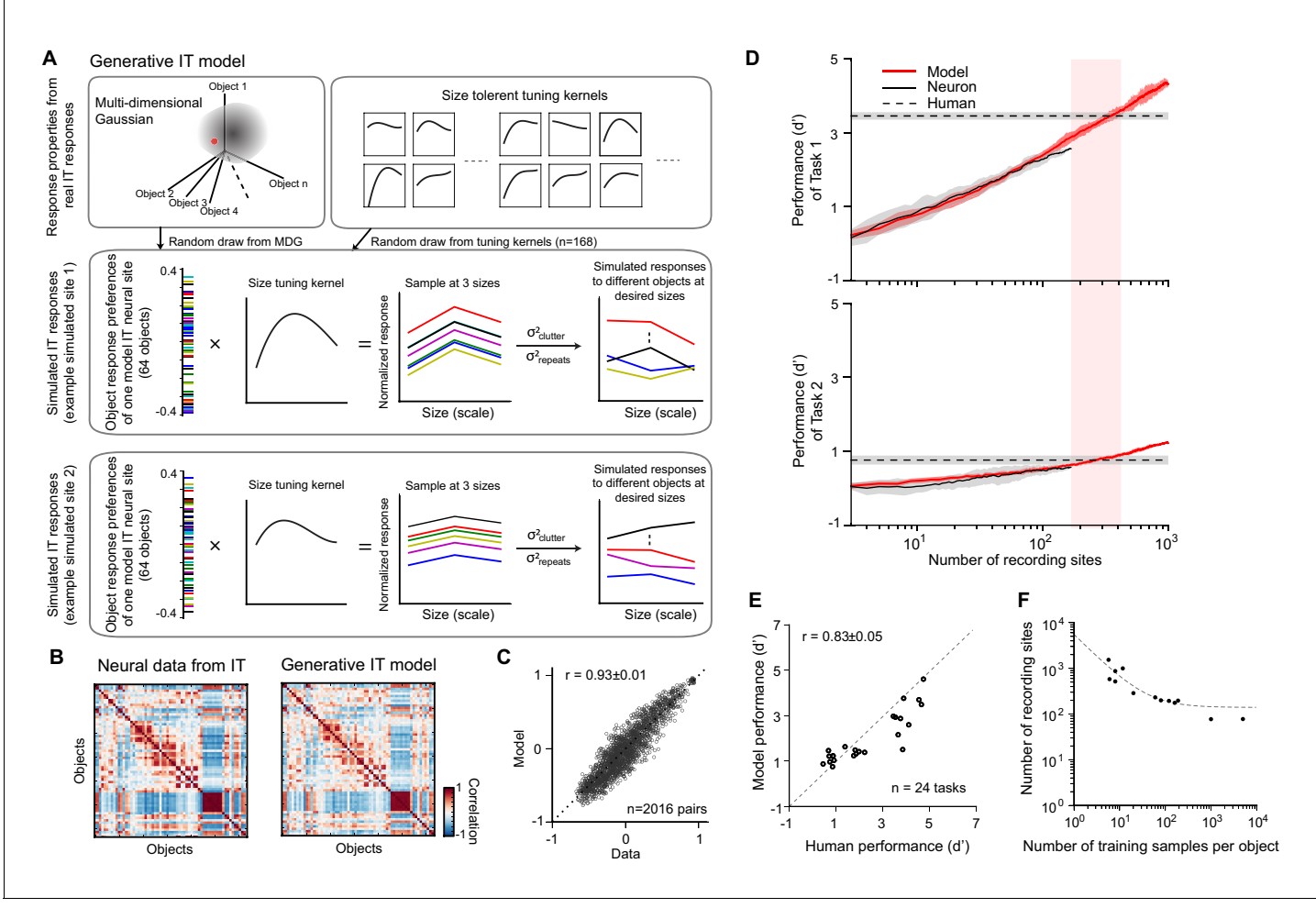

**Figure 3.** Generative ITmodel and validation of the IT-to-core object recognition (COR)-behavior-linking model. (**A**) Generative IT model based on real IT population responses. Top left box: schematic illustration of the neuronal representation space of IT population with a multi-dimensional Gaussian (MDG) model. Each point of the Gaussian cloud is one IT neural site. Middle box: an example of simulated IT neural site. The distribution of object preference for all 64 objects is created by a sample randomly drawn from the MDG (highlighted as a red dot; each color indicates a different object). Then, a size-tuning kernel is randomly drawn from a pool of size-tuning curves (upper right box; kernels fit to real IT data) and multiplied by the object response distribution (outer product), resulting in a fully size-tolerant (i.e., separable) neural response matrix (64 objects × 3 sizes). To simulate the final mean response to individual images with different backgrounds, we added a 'clutter' term to each element of the response matrix ($\sigma^2_{clutter}$; see Materials and methods). To simulate the trial-by-trial 'noise' in the response trials, we added a repetition variance ($\sigma^2_{repeats}$; see Materials and methods). Bottom box: another example of simulated IT site. (**B**) Response distance matrices for neuronal responses from real IT neuronal activity (n = 168 sites) and one simulated IT population (n = 168 model sites) generated from the model. Each matrix element is the distance of the population response between pairs of objects as measured by Pearson correlation (64 objects, 2016 pairs). (**C**) Similarity of the model IT response distance matrix to the actual IT response distance matrix. Each dot represents the unique values of the two matrices (n = 2016 object pairs), calculated for the real IT population sample and the model IT population sample (r = 0.93 ± 0.01). (**D**) Determination of the two hyperparameters of the IT-to-behavior-linking model. Each panel shows performance (d') as a function of number of recording sites (training images fixed at m = 20) for model (red) and real IT responses (black) for two object discrimination tasks (task 1 is easy, human pre-exposure d' is ~3.5; task 2 is hard, human pre-exposure d' is ~0.8; indicated by dashed lines). In both tasks, the number of IT neural sites for the IT-to-behavior decoder to match human performance is very similar (n ~ 260 sites), and this was also true for all 24 tasks (see **E**), demonstrating that a single set of hyperparameters (m = 20, n = 260) could explain human pre-exposed performance over all 24 tasks (as previously reported by *Majaj et al., 2015*). (**E**) Consistency between human performance and model IT-based performance of 24 different tasks for a given pair of parameters (number of training samples m = 20 and number of recording sites n = 260). The consistency between model prediction and human performance is 0.83 ± 0.05 (Pearson correlation ± SEM). (**F**) Manifold of the two hyperparameters (number of recording sites and number of training images) where each such pairs (each dot on the plot) yields IT-based performance that matches initial (i.e., pre-exposure) human performance (i.e., each pair yields a high consistency match between IT model readout and human behavior, as in **E**). The dashed line is an exponential fit to those dots at any of the three sizes as the outer product of the object and size-tuning curves (**A**, bottom). However, since most measured size-tuning curves are not perfectly separable across objects (*DiCarlo et al., 2012*; *Rust and Dicarlo, 2010*) and because the tested conditions included arbitrary background for each condition, we introduced independent clutter variance caused by backgrounds

*Figure 3 continued on next page*

*Figure 3 continued*

on top of this for each size of an object (**A**) by randomly drawing from the distribution of variance across different image exemplars for each object. We then introduced trial-wise variance for each image based on the distribution of trial-wise variance of the recorded IT neural population (***Figure 3—figure supplement 1E***). In sum, this model can generate a new, statistically typical pattern of IT response over a population of any desired number of simulated IT neural sites to different image exemplars within the representation space of 64 base objects at a range of sizes (here targeting 'small,' 'medium,' and 'big' sizes to be consistent with human behavioral tasks; see Materials and methods for details). The simulated IT population responses were all constrained by recorded IT population statistics (***Figure 3—figure supplement 1***). These statistics define the initial simulated IT population response patterns, and thus they ultimately influence the predicted unsupervised neural plasticity effects and the predicted behavioral consequences of those neural effects.

The online version of this article includes the following figure supplement(s) for figure 3:

**Figure supplement 1.** Supplemental information for generative IT model.

neural sites, indicating that our generative IT model has captured the relevant components of the IT population response.

We next set out to choose the two free hyperparameters (number of sites and number of training examples). The crossing point with human performance in ***Figure 3D*** reflects how many neural sites are necessary to reach human performance level for a given number of training samples. Unlike the real IT neural data (n = 168 recording sites) that required extrapolation to estimate the number of sites matching human absolute performance (***Majaj et al., 2015***), we simulated up to 1000 IT sites with the generative model to cover the range of neural sites necessary to reach human performance.

Consistent with ***Majaj et al., 2015***, we found that the number of simulated IT sites required to match human was similar across different tasks (260 ± 23) IT sites given 20 training images (tested over 24 object discrimination tasks: low variation eight-way tests: eight basic level, eight car identification, and eight face identification tasks; previously used in ***Majaj et al., 2015***). Specifically, we here used 260 sites with 20 training samples for all tasks, and the match between the decoded simulated IT performance and human performance over all discrimination tasks was r = 0.83 ± 0.05 (n = 24 tasks), similar to previously reported match between decoded neural IT performance and human for the same tasks (r = 0.868 from ***Majaj et al., 2015***). Note that other specific combinations of the number of IT sites and the number of training examples are also suitable (***Figure 3F***), and we explore this later.

In sum, by setting the two decoder hyperparameters to match initial human performance, we established a fixed linear decoder rule that could be applied to our simulated IT population to quantitatively predict the expected performance of the subject (i.e., the owner of that IT population) for any object discrimination task.

The consequence is that, because the linkage model between the IT population and behavior is now fixed in the model, any changes in the model IT population are automatically mapped to predicted changes (if any) in behavioral performance. From here on, we locked down the generative IT model and the decoders that matched human initial performance (before learning), and combine both of these models later to make predictions of direction and magnitude of behavioral performance change (if any) that should result from any given change in the IT population driven by unsupervised plasticity (***Figure 2***).

## Unsupervised IT plasticity rule

To model the IT neural population response changes that result from the unsupervised visual experience provided to the human subjects, we developed an unsupervised IT plasticity rule guided by previous studies of IT plasticity effects in the rhesus monkey that used the same paradigm of unsupervised visual experience that we provided here to our human subjects (***Li et al., 2009***; ***Li and DiCarlo, 2008***; ***Li and DiCarlo, 2010***). In particular, we set out to build an unsupervised IT plasticity rule that could predict the (mean) response change that occurs in each and every IT neuron as a result of each presented pair of temporally contiguous visual images. We assumed that the same model would also apply to human 'IT' without any parameter modifications (see Discussion).

Those prior monkey studies revealed that exposure to altered ('swapped') visual statistics typically disrupts the size tolerance of single IT neurons, while exposure to normal statistics in visual experience (non-swapped condition) typically builds size tolerance (***Li and DiCarlo, 2010***). To develop our unsupervised IT plasticity rule, we replicated the exact same experiment used in the monkeys on

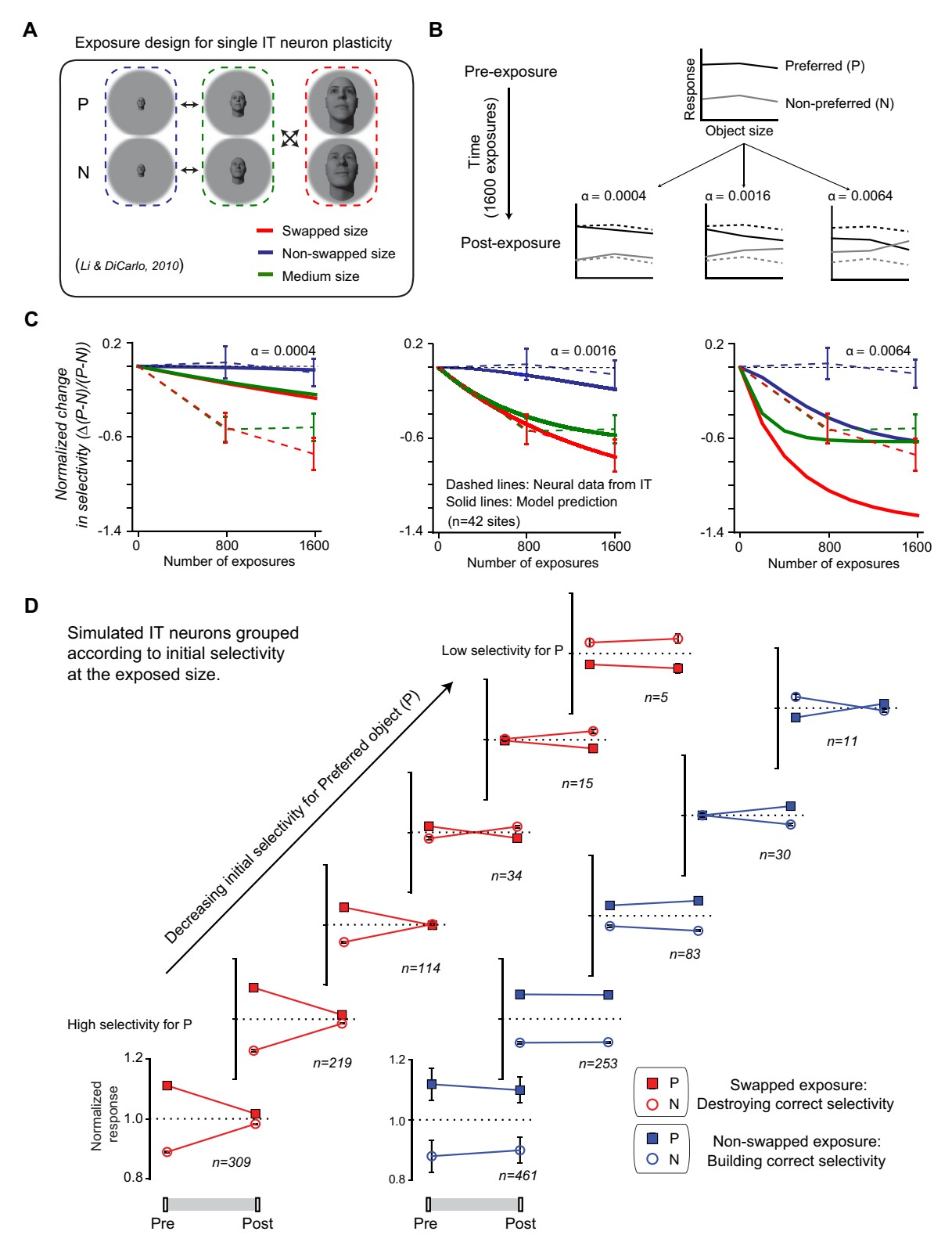

**Figure 4.** Temporal continuity-based inferior temporal (IT) neuronal plasticity rule. (**A**) Illustration of exposure design for IT neuronal plasticity (adapted directly from *Li and DiCarlo, 2010*) measured with single electrodes. P refers to preferred object of the IT unit, and N refers to non-preferred object of that unit. (**B**) We developed an IT plasticity rule that modifies the model neuronal response to each image in individual neural sites according to the difference in neuronal response between lagging and leading images for each exposure event (see Materials and methods). The figure shows the

*Figure 4 continued on next page*

*Figure 4 continued*

model-predicted plasticity effects for a standard, size-tolerance IT neural site and 1600 exposure events (using the same exposure design as *Li and DiCarlo, 2010*; i.e., 400 exposure events delivered [interleaved]) for each of the four black arrows in panel (**A**) for three different plasticity rates. Dashed lines indicate model selectivity pattern before learning for comparison. (**C**) Normalized change over time for modeled IT selectivity for three different plasticity rates. Dashed lines are the mean neuronal plasticity results from the same neural sites in *Li and DiCarlo, 2010* (mean change in P vs. N responses, where the mean is taken over all p > N selective units that were sampled and then tested; see *Li and DiCarlo, 2010*). Solid lines are the mean predicted neuronal plasticity for the mean IT model 'neural sites' (where these sites were sampled and tested in a manner analogous to *Li and DiCarlo, 2010*; see Materials and methods). Blue line indicates the change in P vs. N selectivity at the non-swapped size, green indicates change in selectivity at the medium size, and red indicates change in selectivity at the swapped size. Error bars indicate standard error of the mean. (**D**) Mean swapped object (red) and non-swapped object (blue) plasticity that results for different model IT neuronal sub-groups – each selected according to their initial pattern of P vs. N selectivity (analogous to the same neural sub-group selection done by *Li and DiCarlo, 2010*; c.f. their Figure 6).

simulated IT neural sites. *Figure 4A* illustrates the exposure design for single IT sites, where the preferred object (P) and non-preferred object (N) of each neural site are defined by the magnitude of neuronal activity (z-scored across all objects for each IT site). Selectivity of a neural site is measured by the difference of neuronal responses to its preferred and non-preferred objects (P – N)/(p + N), the same as *Li and DiCarlo, 2010*.

We used a Hebbian-like (associative) plasticity rule (*Caporale and Dan, 2008*; *Hebb, 1949*; *Oja, 1982*; *Paulsen and Sejnowski, 2000*), which updates FR for each pair of images based on the difference of neuronal FR between the lagging and leading images (see Materials and methods). Our plasticity rule states that the modification of FR of each IT unit to the leading image at time t is equal to the difference of FR between lagging and leading images multiplied by a plasticity rate α. This plasticity rule tends to reduce the difference in neuronal responses to consecutive images and implies a temporal association to images presented close in time. The plasticity rule is conceptually similar to previously proposed temporal continuity plasticity or a.k.a. slow feature analysis (*Berkes and Wiskott, 2005*; *Földiák, 1990*; *Földiák, 1991*; *Mitchison, 1991*; *Sprekeler et al., 2007*). It is physiologically attractive because the findings on short-term synaptic plasticity revealed that synaptic efficacy changes over time in a way that reflects the history of presynaptic activity (*Markram et al., 2012*; *Markram et al., 1997*). Even though conceptually similar, our plasticity rule is a 'descriptive' rather than a 'mechanistic' rule of plasticity at all stages of the ventral stream. That is, the rule does not imply that all the underlying plasticity is in IT cortex itself – but only aims to quantitatively capture and predict the changes in IT responses resulting from unsupervised visual experience. It is general in a sense that it can make predictions for different objects or dimensions of variations, but it is (currently) limited in that it only applies to temporally paired image associations, ignores any correlation in the neural response patterns, and assumes that updates occur only in the responses to the exposed images (i.e., non-exposed object/size combinations are not affected).

To show the effects of this unsupervised IT plasticity rule, we illustrate with an example simulated IT neural site. The simulated neural site in *Figure 4B* was initialized to be – like many adult monkey IT neurons – highly size tolerant: its response to a preferred object (P) is always greater than response to a non-preferred object (N) at each size. After applying the unsupervised exposure design in *Figure 4A* (200 exposure events for each arrow, 1600 exposure events in total), the responses to each of the six conditions (2 objects × 3 sizes) evolved as shown in *Figure 4B*. We note two important consequences of this plasticity rule. First, because the rule was designed to decrease the difference in response across time, responses to images presented consecutively tend to become more similar to each other, which results in a reduction in the response difference between P and N at both the swapped and the non-swapped sizes. Second, once the neural site reached a state in which its response is no different over consecutively exposed images, the learning effect saturates. Notably, unlike adaptive changes in plasticity rate in the typical supervised optimization of deep neural networks (*Kingma and Ba, 2014*), our plasticity rate is kept constant over the 'lifetime' of the model. The gradual shrinkage of learning effect (Δ(P – N)/(p + N)) as more and more exposure events are provided was a consequence of the gradual reduction in the absolute difference between neuronal responses to the two consecutive images that makeup each exposure event.

There is only one free parameter in our plasticity rule equation – the plasticity rate α. We determined this parameter using the single-electrode physiology data collected previously in the lab (*Li and DiCarlo, 2010*). *Figure 4C* shows the average IT plasticity effect that results from different

settings of $\alpha$ (here the plasticity effect is defined by the normalized changes in selectivity: $\Delta(P-N)/(P-N)$, exactly as was done in *Li and DiCarlo, 2010*). As expected, a higher plasticity rate ($\alpha$) results in greater model IT plasticity effects (*Figure 4C*). We chose the plasticity rate ($\alpha$) that best matched the prior monkey IT neurophysiology results (i.e., the $\alpha$ that resulted in the minimal difference between the model IT plasticity effect [solid lines] and the experimentally reported IT plasticity effect [dashed lines] for swapped, non-swapped, and medium object sizes; see *Figure 4C* middle). The best $\alpha$ is 0.0016 nru per exposure event (nru = normalized response units; see Materials and methods for intuition about approximate spike rate changes). Once we set the plasticity rate, we locked it down for the rest of this study (otherwise noted later where we test rate impact).

We next asked if our IT plasticity rule naturally captured the other IT plasticity effects reported in the monkey studies (*Li and DiCarlo, 2010*). Specifically, it was reported that, for each neural site, the selectivity that results from a fixed amount of unsupervised exposure depends on the initial selectivity of that site. Thus, the unsupervised 'swapped' experience manipulation causes a reduction of selectivity for neural sites that show a moderate level of initial P (preferred object) vs. N (non-preferred object) selectivity at the swapped size, and the same amount of unsupervised experience *reverses* the selectivity of neuronal sites that show a low level of initial selectivity at the swapped size (i.e., cause the site to, oxymoronically, prefer object N over object P). *Li and DiCarlo, 2010* also reported that the more natural, 'non-swapped' experience manipulation caused a *building* of new selectivity (for neuronal units that initially show a strong preference for P at some sizes, but happened to have low P vs. N selectivity at the non-swapped size).

We tested for both of these effects in our model by selecting subsets of neural sites in the simulated IT population in exactly the same way as *Li and DiCarlo, 2010* (sampled from n = 1000 simulated IT units) and then applied the plasticity rule to those units. We found a very similar dependency of the IT plasticity to those previously reported IT plasticity effects (*Figure 4D*; cf. see Figures 6 and 7 of *Li and DiCarlo, 2010*).

Given that our IT plasticity rule tends to pull the response of temporally contiguous images toward each other (*Berkes and Wiskott, 2005*; *Földiák, 1990*; *Földiák, 1991*; *Mitchison, 1991*; *Sprekeler et al., 2007*), it is not entirely obvious how this can build selectivity (i.e., pull response to P and N apart). The reason this occurs is that some IT neural sites have (by chance draw from the generative model of IT, above) initially high selectivity for P vs. N at the medium size and no selectivity at (e.g.) the big size. (Indeed, such variation in the IT population exists as reported in *Li and DiCarlo, 2010*.) By design, the non-swapped ('natural') unsupervised exposure temporally links $P_{med}$ (high response) with $P_{big}$, which – given the plasticity rule – tends to pull the $P_{big}$ response upward (pull it up higher than $N_{big}$). In addition, the non-swapped exposure links $N_{med}$ (low response) with $N_{big}$, which can pull the $N_{big}$ response downward (provided that the $N_{med}$ response is initially lower than the $N_{big}$ response). Both effects thus tend to increase the $P_{big}$ vs. $N_{big}$ response difference (i.e., both effects tend to 'build' selectivity for P vs. N at the big presentation size, which results in the neural site preferring object P over object N at both the medium and the big sizes – a property referred to as size 'tolerance'). This effect is observed in single IT neural site size-tuning curve for P and N before and after learning (see Figure 3 in *Li and DiCarlo, 2010*). Indeed, it is this effect that conceptually motivated temporal contiguity plasticity in the first place – natural-occurring statistics can be used to equalize the responses to the same object over nuisance variables (such as size).

In sum, our very simple IT plasticity rule quantitatively captures the average IT plasticity effects for which its only free parameter was tuned, and it also naturally captures the more subtle IT neural changes that have been previously described.

## Putting together the overall model to predict human unsupervised learning effects

To summarize, we have (1) built and tested a generative IT model that captured the object representation space and variability in the actual primate IT population; (2) locked down a set of parameters of a linear decoder rule that quantitatively links the current state of the simulated IT population to initial human performance on any discrimination task (including the ones we plan to test); and (3) defined an IT plasticity rule that describes how each individual IT neural site changes as a result of each unsupervised exposure event, and we locked down the only free parameter (plasticity rate) in that rule to match existing monkey IT plasticity data (see *Figure 1—figure supplement 1A*). At this

point, we could – without any further parameter tuning – combine each of these three model components into a single overall model that predicts the direction, magnitude, and time course of human unsupervised learning effects that should result from any unsupervised learning experiment using this exposure paradigm (pairwise temporal image statistics).

Specifically, to generate the predictions for each of unsupervised learning experiments (u: u1, u2, u3; see *Figure 2*), we (1) initialized a potential adult human IT (from the generative IT model) with a total of 260 simulated IT recording sites; (2) built linear decoders for the planned object discrimination tasks that read from all 260 sites, using 20 training examples for each and every task; (3) froze the parameters of all such decoders (i.e., froze the resulting weighting on each simulated IT neural site on the 'subject's' object choice decision); (4) 'exposed' the IT model population to the same unsupervised temporal exposure history as the human subjects, using the IT plasticity rule to update the model 'IT' after each temporally adjacent image exposure pair to update the responses of each simulated IT neural site (note that the plasticity trajectory of each neural site is dependent on both its initial object/size response matrix [1], and the sequence of images applied during unsupervised experience [u]); (5) measured the changes in 'behavioral' performance of the overall model (changes in object discrimination performance of the [frozen] decoders [2]); and (6) took those changes as the model predictions of the changes in human performance that should result from that unsupervised experience (u). Again we emphasize that, while the overall model relies heavily on data and parameters derived explicitly or implicitly from these prior studies (*Li and DiCarlo, 2010*; *Majaj et al., 2015*), no components or parameters of this model nor its predictions depended on the behavioral data collected in this study.

To give robust model estimates of the average predicted behavioral effects, we repeated this process (1–6) 100 times for each experiment (u) and averaged the results, which is analogous to running multiple subjects and averaging their results (as we did with the human data; see *Figure 2*). For clarity, we note that the prediction stochasticity is due to random sampling of the IT generative population, the clutter variability introduced in the generative IT model when generating the initial population response for each test image, the trial-by-trial variation in the simulated IT responses, the random unsupervised exposure event sequence (see Materials and methods), and randomly drawn test images, all of which we expect to average out.

Note that, in expecting that these overall model predictions might be accurate, we are implicitly making the following assumptions: (1) monkey IT and human IT are approximately identical (*Kriegeskorte et al., 2008*; *Rajalingham et al., 2015*), (2) the linkage of IT to behavioral performance is approximately identical (as suggested by *Majaj et al., 2015*), (3) human IT unsupervised plasticity is the same as monkey IT unsupervised plasticity, and (4) humans do not re-learn or otherwise alter the assumed mechanistic linkage between IT and behavior during or after unsupervised visual experience (at least not at the time scales of these experiments: 1.5–2 hr).

## Results: predicted learning vs. observed learning

*Figure 5A, D, E* show the model-predicted learning effects (black solid line) for each of the three unsupervised experiments (u1, u2, u3) plotted on top of the observed measured human learning effects (red line, reproduced from the learning effects shown in *Figure 2* bottom). For each experiment, we found that the overall model did a very good job of predicting the direction, magnitude, and time course of the changes in human behavior. The directional predictions are not surprising given prior qualitative results, but the accurate predictions of the magnitude and time course are highly non-trivial (see below). Despite these prediction successes, we also noticed that the predictions were not perfect, most notably after large numbers of unsupervised exposures (e.g. *Figure 5E*, rightmost points), suggesting that one or more of our assumptions and corresponding model components are not entirely accurate (see Discussion).

Given the surprising overall quantitative accuracy of the model predictions straight 'out of the box,' we wondered if those predictions might somehow occur even for models that we had not carefully tuned to the initial (pre-exposure) human performance and the previously reported IT plasticity. That is, which components of the model are critical to this predictive success? We tested this in two ways (focusing here on experiment u1).

First, we built model variants in which the IT plasticity rate (α) was either four times smaller or four times bigger than empirically observed in the prior monkey IT neurophysiology (solid gray lines) and re-ran the entire simulation procedure (above). In both cases, the predictions of these (non-

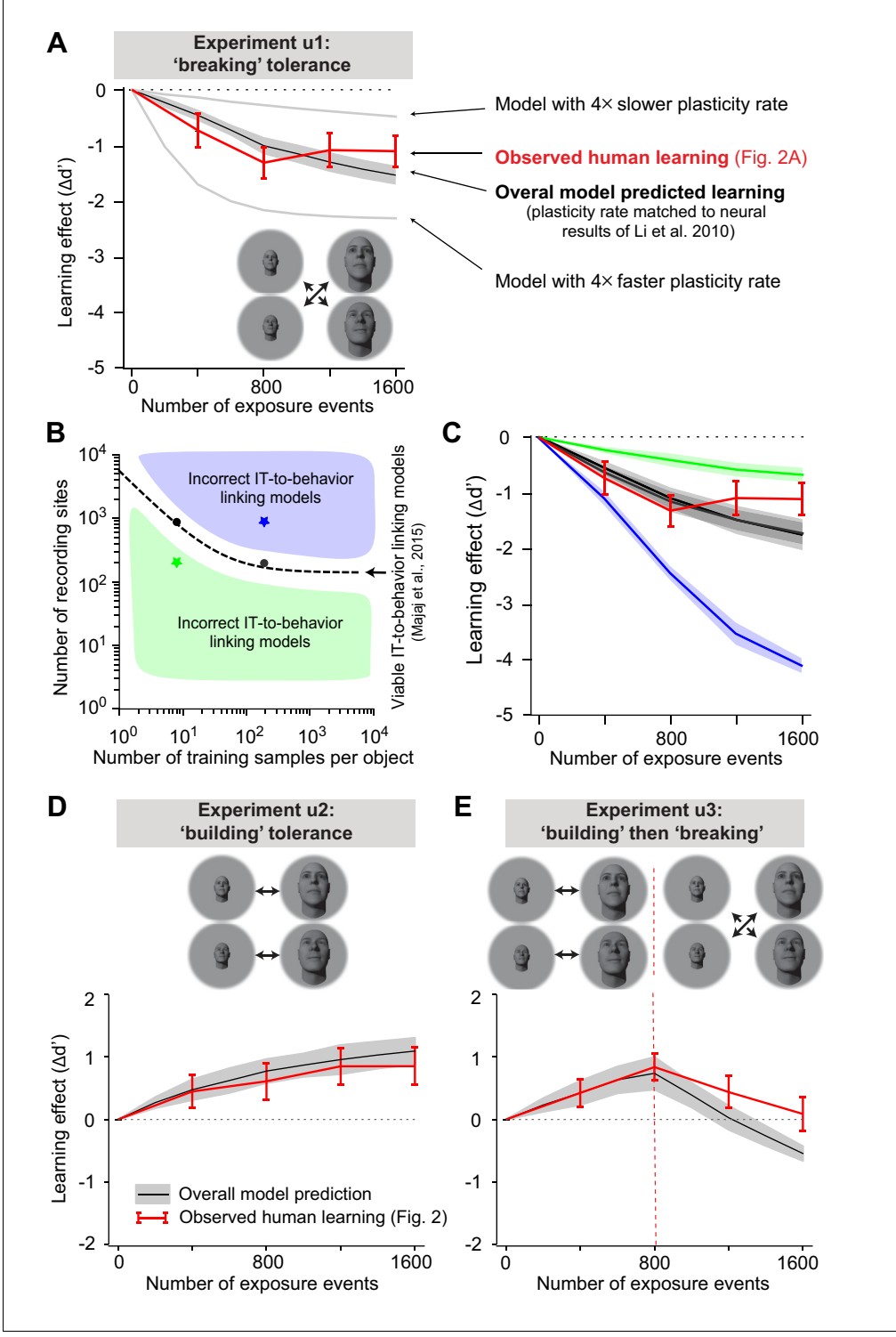

**Figure 5.** Overall model-predicted learning effects vs. actual learning effects. (**A**) Overall model-predicted learning effect (solid black line) for experiment u1 (swapped exposure) with the IT-to-behavior-linking model matched to initial human performance (hyperparameters: number of training images m = 20, number of model neural sites n = 260; see **Figure 3**) and the IT plasticity rate matched to prior IT plasticity data (0.0016; see **Figure 4**). Red line indicates measured human learning effect (reproduced from **Figure 2A**, lower). Gray lines indicate model predictions for four times smaller plasticity rate and four times larger plasticity rate. Error bars are standard error over 100 runs of the overall model; see text. (**B**) Decoder hyperparameter space: number of training

*Figure 5 continued on next page*

*Figure 5 continued*

samples and number of neural features (recording sites). The dashed line indicates pairs of hyperparameters that give rise to IT-to-behavior performances that closely approximate human initial (pre-exposure) human object recognition performance over all tasks. (C) Predicted unsupervised learning effects with different choices of hyperparameters (in all cases, the IT plasticity rate was 0.0016 – i.e., matched to the prior IT plasticity data; see *Figure 4*). The two black lines (nearly identical, and thus appear as one line) are the overall model-predicted learning that results from hyperparameters indicated by the black dots (i.e., two possible correct settings of the decoder portion of the overall model, as previously established by *Majaj et al., 2015*). Green and blue lines are the overall model predictions that result from hyperparameters that do not match human initial performance (i.e., non-viable IT-to-behavior-linking models). (D) Predicted learning effect (black line) and measured human learning effect (red) for building size-tolerance exposure. (E) Model-predicted learning effect (black line) and measured human learning effect (red) for building and then breaking size-tolerance exposure. In both (D) and (E), the overall model used the same parameters as in (A) (i.e., IT plasticity rate of 0.0016, number of training samples m = 20, and number of model neural sites n = 260).

biology-matched) model variants were now clearly different in magnitude than the observations (*Figure 5A*). This result is arguably the strongest evidence that the single-unit IT plasticity effects fully account for – and do not over-account for – the human unsupervised learning effects presented thus far.

Second, we built model variants in which the two decoder hyperparameters (number of neural sites and number of training images) were no longer correctly aligned with the initial human performance levels. *Figure 5B* illustrates the two-dimensional hyperparameter space, and the dashed line represents potential choices of the two hyperparameters that match human initial performance (the IT-to-COR-behavior-matching manifold; *Figure 3F*). Regions above (or below) that manifold indicate hyperparameter choices where the decoders are better (or worse) performing than initial human performance. We found that the unsupervised learning effects predicted by the overall model (*Figure 5C*, two black lines on top of each other corresponding to two choices of hyperparameters, black dots in *Figure 5B*) continued to well-approximate human learning effects. This was also true for other combinations of hyperparameters along the dashed black manifold in *Figure 5B* (~10 combinations tested; results were similar to those shown in *Figure 5C–E*, not shown). In other words, for model settings in which the model variant was in line with the biological initial state, the predictions of the unsupervised learning effects remained similarly accurate. This is a nice robustness check on the model simulations and predictions. (However, as a side note orthogonal to our goals here, this result also means that, as in prior work [*Majaj et al., 2015*], we cannot use this analysis to determine which of these model variants is more matched to the biology.)

In contrast, when we built model variants in which the choices of the two hyperparameters did not match human initial performance, the unsupervised learning effect predicted by the overall model clearly differed from the observed human learning effect. Specifically, when an overall model starts off with 'super-human' performance, it overpredicted the learning effect; and when a different model starts off as 'sub-human,' it underpredicted the learning effect.

In sum, it is not the case that any model of the form we have built here will produce the correct predictions – proper (biological) setting of the unsupervised IT plasticity rate and proper (biological) setting of the IT-to-COR-behavior-linkage model are both critical. It is important to note that we did *not* tune these biologically constrained hyperparameters based on fitting the unsupervised behavioral learning effects in *Figure 2* – they were derived in accordance with prior neurobiological work as outlined above.

## The unsupervised learning effect depended on the initial task difficulty

So far, we have established a quantitative overall model that quite accurately predicted the direction, magnitude, and time course of learning effects resulting from a range of unsupervised exposure manipulations. For each of those tests, we focused on object discrimination tasks that had an intermediate level of initial task difficulty (face discrimination tasks with initial d' around 2.0), so that we had dynamic range to see both increases and decreases in performance (e.g., *Figure 2*). However, we noticed that our IT plasticity rule seemed to imply that those learning effects would depend on the strength of the initial selectivity of individual IT neural sites for the exposed objects (i.e., the

initial P vs. N response difference). The intuition is that this response difference is the driving force for IT plasticity updates (e.g., no difference leads to no update, large difference leads to large update). This in turn implied that the learning effect size should depend on the initial task performance (d').

To test for this dependence, we focused on the unsupervised size tolerance 'breaking' manipulation (as in u1, *Figure 2A*, but with 800 unsupervised 'swapped' exposures; see Materials and methods) and tested new sets of human subjects using a wide range of initial task difficulties, ranging from subordinate object discriminations (low d') to basic-level object discriminations (high d'). We focused on 13 size-specific object discrimination sub-tasks with either small-medium-size swapping exposure or medium-big-size swapping exposure. Each subject received only one exposure variant (see Materials and methods). For each exposure variant, 20–40 new human subjects were tested, and we quantified the unsupervised learning effect ('breaking') as the change (from initial) in performance (relative to control objects, as in *Figure 2A*).

*Figure 6B* shows that unsupervised learning effect plotted against pre-exposure task difficulty for all 13 object discrimination tasks. This result not only confirms that this unsupervised learning effect is observed for a range of object discriminations (e.g., not just face objects), but it also showed a relationship between task difficulty (d') and the magnitude of that learning effect. In particular, for initially easy tasks (d' > ~2.5) and initially difficult tasks (d' < ~0.5), we observed a smaller learning effect than tasks with intermediate initial performance.

We found that our overall model quite naturally – for the reasons outlined above – predicted the smaller learning effect for initially difficult tasks (the left side of *Figure 6B*). Notably, the model as defined above did not naturally predict the lack of observed learning effects for the initially easy tasks (high initial d') – it tended to overpredict the magnitude of behavioral changes that will result

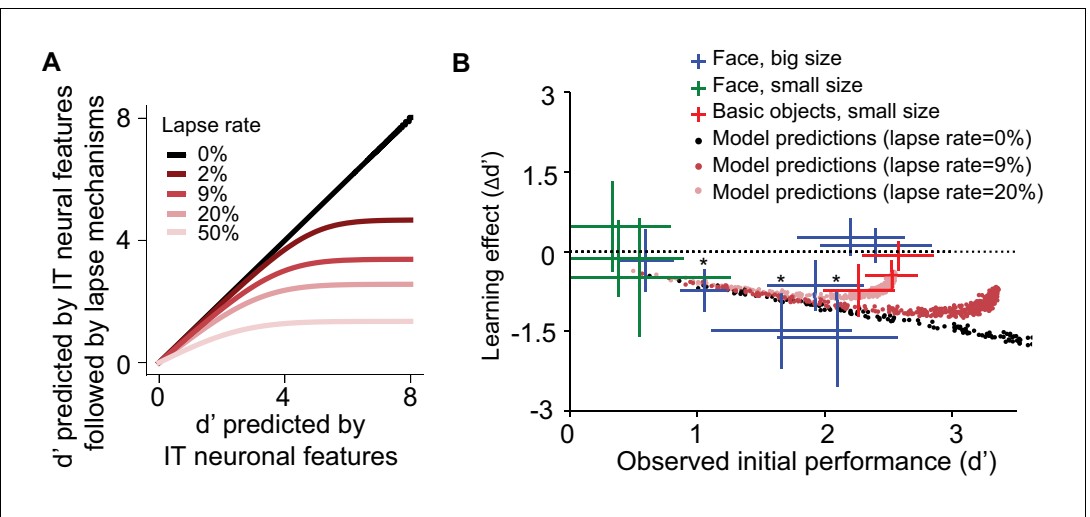

**Figure 6.** Learning effect as a function of initial task difficulty. (**A**) Illustration of the saturation of measured d' that results from the assumption that the subject guesses on a fraction of trials (lapse rate), regardless of the quality of the sensory evidence provided by the visually evoked inferior temporal (IT) neural population response (x-axis). (**B**) Measured human learning effect for different tasks (colored crosses) as a function of initial (pre-exposure) task difficulty (d') with comparison to model predictions with or without lapse rate (dots). Each cross or dot is a specific discrimination task. For crosses, different colors indicate different types of tasks and exposures: green indicates small-size face discrimination learning effect induced with medium-small swapped exposure (n = 100 subjects); blue indicates big-size face discrimination learning effect induced with medium-big swapped exposure (n = 161 subjects); red indicates small-size basic-level discrimination learning effect induced with medium-small swapped exposure (n = 70 subjects). Performance is based on population pooled d'. Error bars indicate bootstrapped standard error of the mean population pooled d' (bootstrapping is performed by sampling with replacement across all trials). p-value is directly estimated from the bootstrapped distributions of performance change by comparing to no change condition. * indicates p-value<0.05.

The online version of this article includes the following figure supplement(s) for figure 6:

**Figure supplement 1.** Distribution of initial task performance.

in those high d' task (see *Figure 6B*, black dots). However, we realized that, if we assumed that the model also has a lapse rate similar to that of humans (*Prins, 2012*), then this discrepancy might go away. That is, we assume that there is some non-zero fraction of trials for which the subject guesses or makes motor errors, regardless of the quality of the sensory-driven information. The intuition here is that human subjects make task-independent mistakes ('lapses'), and even a low rate of random lapses puts a ceiling on the d' value that can be experimentally measured (*Figure 6A*). In the context of our learning experiments, this assumption would mean that the underlying neural representation might indeed be changing a great deal (at least that is what our current model predicts), but those changes cannot be measured as changes in human performance in the face of a lapse-rate induced measurement ceiling (e.g., a sensory evidence d' of 5 could change to a sensory evidence d' of 3.5 [a large d' change of 1.5], but we would measure an observed behavioral d' of ~3 in both cases and thus report a behavioral d' change of ~0). In contrast, the overall model that we described above had a zero lapse rate, which meant that we could measure changes in its behavioral performance for even very large initial d' values.

To explore this, we asked: what is the (mean) lapse rate of the human subjects in our experiments? To estimate this, we used half our human data to rank the average initial human performance of each task which we take as an estimate of the ordering of those tasks in terms of available sensory evidence. We then used the remaining data to plot the observed human performance on each task (*Figure 6—figure supplement 1A*). We found that the average performance tended to plateau around 90%, which we take as an indication of a performance ceiling that cannot be explained by a lack of sensory evidence, and thus we attribute to multiple types of downstream errors collectively referred to as lapses. We also directly quantified the distribution of task performance accuracy for basic-level tasks (easy tasks: d' > 2.5) in our experiments (*Figure 6*) and found that the distribution has a maximum of 95.0% (*Figure 6—figure supplement 1B*). These analyses suggest that the lapse rate of our subject pool is ~10% (i.e., 95% accuracy for two choice tasks with perfect sensory evidence), which is consistent with prior work on human lapse rates (<20%; *Manning et al., 2018*). We simulated the effect of 9% lapse rate and 20% lapse rate (i.e., we told the model to make random guesses on 9% or 20% of trials, regardless of the strength of sensory evidence) and found that these new overall models reasonably explained the dependence of the observed magnitude of human d' changes as a function of initial human d' (*Figure 6B*).

In sum, we interpret the lapse rate analyses not as a failure of the overall model, but instead as a limitation of our psychophysical experiments in this study. That is, the lapse rate estimate is consistent with the hypotheses that, in the high initial d' range, the IT population is changing (indeed, the exposure conditions are close to the conditions of the original monkey neural experiments; *Li and DiCarlo, 2010*), but that, in the face of a lapse rate, the behavioral consequences of those changes are predicted to be small relative to the effects of downstream biological variability. That being said, it also means that the current study is simply not able to test the IT plasticity to behavioral-learning linkage in the initial high d' range, and we take that as a target for future experimental work (see Discussion).

## Discussion

The goal of this study was to ask if previously reported temporal contiguity-driven unsupervised plasticity in IT neurons quantitatively accounts for temporal contiguity-driven unsupervised learning effects in humans.

To do that, we built an overall computational model to predict human performance change resulting from plasticity in individual IT neural site FRs under the paradigm of unsupervised temporal contiguity exposure (temporally contiguous pairs of images). The overall model had three core components: (1) a generative model of a baseline adult IT neuronal population, (2) an IT-population-to-COR-behavior-linking model (adopted directly from *Majaj et al., 2015*), and (3) an IT plasticity rule that aimed to capture and predict how pairs of temporally associated images lead to updates in the (future) image-driven responses of each individual IT neural site. Each of these three model components was guided by prior work and constrained by prior data to set its parameters, and the combined overall computational model was directly used to predict human learning effects without further parameter tuning.

To test the overall model, we asked the model to predict the human performance changes for three separate unsupervised learning experiments and compared those predictions with the human performance changes (averaged over human subjects) that we measured in those three experiments. We found that the direction, magnitude, and time course of those mean unsupervised learning effects were all quite well predicted by the overall model (but not perfectly predicted). We also found that the model could naturally explain the dependence of the measured unsupervised learning on initial object discrimination difficulty, but that it could only fully do so when we assumed model 'behavioral' lapse rates that were similar to that estimated in our human subject pool.

In sum, this work establishes a quantitative linking model between the plasticity in individual IT neurons and human behavioral changes (both improvements and disruptions) for temporal contiguity-driven unsupervised learning for the designed tasks. More broadly, the accuracy of the model predictions supports the overarching hypothesis that temporally contiguous unsupervised learning could help shape neural representations that underlie robust (i.e., tolerant) COR, even in adults.

We were somewhat surprised that the overall model did such an accurate job of predicting the human learning effects over time essentially from predicted updates on the responses of IT neural sites. This was surprising because the overall model implicitly assumes that monkey IT and human IT are approximately identical (*Kriegeskorte et al., 2008*; *Rajalingham et al., 2018*), the linkage of IT to COR behavior is approximately identical in monkeys and humans (as previously suggested; *Majaj et al., 2015*), human unsupervised IT plasticity is the same as monkey IT plasticity, and that little or no behaviorally relevant plastic changes occur in the mechanistic linkage between IT and behavior during or after unsupervised visual experience (at least not at the time scales of these experiments: 1.5–2 hr). Of course, the results here do not prove all of the above assumptions to be correct. Indeed, another interpretation of these results is that many of those assumptions are incorrect, but that the errors they each induce in the model predictions coincidentally cancel each other out. However, based on both the prior work cited above and the current study, we believe that it is more parsimonious to assume that the model is predictively accurate because all of the above are approximately correct until further experiments show otherwise.

We think it is quite interesting and demonstrative of this approach that the overall model could predict behavioral learning effects in the low initial d' and moderate initial d' regime for different exposure types (build or break), even though the images used in those tasks are in a different d' regime from the specific images used in the neuronal plasticity experiments used to tune the model learning (i.e., those neural experiments were done with images in a high d' regime). We note that the ability to make such tests in these other d' regimes is the key reason that we built the overall composite model in the first place: a generative IT model (simulating IT population patterns to a wide range of image conditions) + an IT neural-to-behavior-linking model + an IT plasticity model (independent of specific images). End-to-end models such as this allow us to predict effects under conditions that no experiment has yet tested, and to then test some of them as we have done here. That is, the integrated model imparts new generalizable knowledge and understanding.

We noted a small discrepancy between the predictions of the model and the human learning data at the longest exposure durations that we tested (1600 exposure; ~1.5 hr; see *Figure 5A,C*), where the model predicted slightly stronger behavioral changes than measured. The most likely possibility is that learning over long periods of unsupervised exposure involves more complicated neural mechanisms than those that are captured by our simplified unsupervised IT plasticity rule. For example, perhaps the plasticity rate slows down as the subject fatigues in the experiment. Or perhaps the plasticity mechanisms involve some type of renormalization of the responses of each IT neuron to retain some selectivity to different objects, as motivated by prior theoretical work on temporal contiguity learning (*Sprekeler et al., 2007*; *Wiskott and Sejnowski, 2002*). Similarly, plasticity along the ventral stream could involve homeostatic range adjustment, which is fundamental to individual neurons (*Turrigiano and Nelson, 2004*), as motivated by studies of LTP and LTD plasticity in V1 neurons (e.g., BCM rules; *Bienenstock et al., 1982*; *Toyoizumi et al., 2005*). While we did not explicitly model any of these neural plasticity effects, they could be explored in future modeling studies with even tighter tests on neurons and behavior than we achieved here.

The dependency of behavioral learning effects on initial task difficulty is also an important observation. The empirical human results show that tasks with moderate initial difficulty give rise to maximum measurable learning effects in the paradigm used here. The learning effects for hard to

moderate tasks are naturally predicted with the initial overall model. However, for easy tasks (typically d' > ~2.5), our initial overall model predicted a larger learning effect than we empirically observed in humans. In hindsight, this discrepancy seems obvious if we assume that human subjects make errors unrelated to the sensory evidence (e.g., due to inattention or motor error). In prior work, this is quantified as a lapse rate (*Madigan and Williams, 1987*; *Manning et al., 2018*; *Prins, 2012*; *Wichmann and Hill, 2001*). Our simulation, and previous studies, suggests that the existence of lapse rate imposes a ceiling on measured performance (*Gold and Ding, 2013*; *Pisupati et al., 2021*; *Prins, 2012*; *Wichmann and Hill, 2001*) and thus would strongly mask measured performance changes in that (high d') regime. In other words, any performance changes within that ceiling would be virtually undetectable. Even though we were able to indirectly estimate the lapse rate of our subject pool, we did not make direct measurements of lapse rate. Furthermore, the influence of lapse rate on learning could be complex, for example, it can fluctuate across time depending on brain states (*Ashwood et al., 2020*). Therefore, future experiments are needed to try to better estimate unsupervised performance changes in the high d' regime (perhaps with same images as the neural experiments) and comparing with this model and others. This might be done by precisely measuring lapse rate for each human subject to possibly allow extraction of even small accuracy changes within each subject, controlling/minimizing lapse rate in a lab environment, giving larger experimental doses of unsupervised exposure, or all of these. However, our results here cannot rule out the possibility that no behavioral effects will be found in this high initial performance regime, no matter how strong the unsupervised exposure dose. Such a finding would falsify the model presented here.

In this work, since we want to establish a quantitative link between single IT neuronal plasticity and human unsupervised learning effects, we matched many of the conditions to those used in prior IT plasticity work. For example, we used the exact same exposure procedure to introduce temporal continuity-based learning for size tolerance (*Li and DiCarlo, 2010*). However, our study still included several differences compared to prior IT plasticity work. For completeness, we list these differences and our assumptions here. First, since we rely on a previous IT population response dataset (*Majaj et al., 2015*) to create a generative IT model and establish the link between IT population activity and human object recognition performance, we rendered test images from the same 3D objects as *Majaj et al., 2015* with background variation and cover trials to prevent from pixel matching in human object discrimination tasks. We assumed that the IT neurons are not making net unsupervised changes during each test phase in their image preference patterns as a result of the image background or randomly interleaved cover image, and therefore, the net learning effect does not depend on what is shown during each test phase. Second, and related, we assumed that the IT generative model could reasonably approximate the variations in IT responses caused by changes in background clutter. Third, the human discrimination tasks are in the lower d' regime compared to the images used in the prior electrophysiology work. We assumed that the learning rate estimated in the high d' regime for an IT population is generalizable to IT neural plasticity in this regime. Because the model predictions turned out to be surprisingly accurate, we take these differences and the corresponding assumptions (above) as a strength of this study rather than as a weakness, yet the differences do admit the possibility that a fortuitous coincidence in assumption errors led to accurate model predictions rather than our preferred interpretation that the assumptions and the model are all approximately correct. Future neural recordings in non-human primates with behavioral learning measurement could be helpful to directly validate the link suggested by our model using the exact same objects and test images as the ones in our human psychophysics.

Our study tested unsupervised learning effects for multiple types of object discriminations (*Figure 6*). However, because learning effects are easiest to measure in the initial task difficulty regime of subordinate-level discrimination (*Figure 6*), we focused much of the psychophysical testing here on face discrimination tasks. This choice was only for convenience, and none of the results here or in prior psychophysical work (*Cox et al., 2005*; *Wallis and Bülthoff, 2001*) suggest that temporal continuity-induced changes in face discrimination are different than those of any other subordinate object discriminations. However, it is well established that IT neurons that respond more to faces than other objects are strongly spatial-clustered within both the monkey and the human ventral stream (*Kanwisher et al., 1997*; *Tsao et al., 2003*), and those neurons often have responses that could be used to support discrimination behavior among different faces (*Chang and Tsao, 2017*). Do our results imply that IT neurons within these large clusters are more (or less) susceptible to

unsupervised plasticity than other IT neurons with face discrimination signals that are not clustered? No, the results here do not add evidence for or against those hypotheses. Briefly, the IT-to-behavior-linking model used here assumes that each task will be driven by the IT neurons that are most useful for that task, as discovered by linear classifiers (*Majaj et al., 2015*). This linking model does not explicitly care if the IT neurons are clustered or not. The null hypothesis here is that all IT neurons have equal plastic potential, regardless of if they are strongly clustered ('face patch') or not (with the degree of observed plasticity then depending only on the visual input statistics). Alternative hypotheses are that neurons within IT face patches have higher (or lower) plastic potential than other IT face-encoding neurons or other IT neurons in general. Our results are consistent with the null hypothesis (i.e., all IT neurons equally potentially plastic), provided that the average plasticity rate is in line with the average IT plasticity rate observed in the prior monkey studies. More targeted future neurophysiology experiments could test these specific alternative hypotheses.

## Tolerant object recognition and temporal continuity-driven unsupervised learning

Human (and monkey) visual object recognition is highly tolerant to object viewpoint, even under short, but natural, viewing durations of ~200 ms referred to as 'core object recognition' (COR) (*DiCarlo et al., 2012*). Much evidence suggests that this ability derives from neural non-linear processing (and thus neural re-representation) of the incoming image along the ventral visual stream, and some ANN models have become reasonably accurate emulators of that non-linear processing and of its supported COR behavior (*Cadieu et al., 2014*; *Krizhevsky et al., 2017*; *Kubilius et al., 2018*; *Yamins et al., 2014*). However, because the 'learning' of those models is highly non-biological (in the sense that millions of labeled images are used to explicitly supervise the learning), a key question remains completely open: how does the ventral stream develop its non-linear processing strategy?

One proposed idea is that, during postnatal development and continuing into adulthood, naturally occurring temporally continuous visual experience can implicitly instruct plasticity mechanisms along the ventral stream that, working together, lead to the transform-invariant object representation (*Berkes and Wiskott, 2005*; *Einhäuser et al., 2005*; *Földiák, 1991*; *Wallis et al., 2009*). Intuitively, the physics of time and space in our natural world constrains the visual experience we gain in everyday life. Because identity-preserving retinal projections often occur closely in time, the spatiotemporal continuity of our viewing experience could thus be useful to instructing the non-linear processing that in turn supports highly view-tolerant object recognition behavior. Under this hypothesis, objects do not need to be labeled per se, they are simply the sources that statistically 'travel together' over time.

We are not the first to propose this overarching hypothesis or variants of it as the theoretical idea dates back to at least ~1960 (*Attneave, 1954*; *Barlow, 1961*). Földiák suggested that the internal representation should mimic physical entities in real life, which are subject to continuous changes in time (*Földiák, 1990*; *Földiák, 1991*). This process is purely unsupervised and achieves transformation invariance by extracting slow features from quickly varying sensory inputs (*Berkes and Wiskott, 2005*; *Sprekeler et al., 2007*; *Wiskott and Sejnowski, 2002*). A range of mathematical implementations of learning rules (*Berkes and Wiskott, 2005*; *Földiák, 1991*; *Isik et al., 2012*; *Körding et al., 2004*; *Wiskott and Sejnowski, 2002*) all include variants of this same conceptual idea: to achieve response stability of each neuron over time (while also maintaining response variance over the full population of neurons). Various synaptic plasticity mathematical rules and associated empirical observations support this form of unsupervised learning: Hebbian learning (*Hebb, 1949*; *Földiák, 1991*; *Löwel and Singer, 1992*; *Paulsen and Sejnowski, 2000*), anti-Hebbian learning (*Földiák, 1990*; *Mitchison, 1991*; *Pehlevan et al., 2017*), BCM rule (*Bienenstock et al., 1982*; *Toyoizumi et al., 2005*), and spike-timing-dependent plasticity (*Caporale and Dan, 2008*; *Markram et al., 1997*; *Rao and Sejnowski, 2001*). This prior work showed that unsupervised learning of neural representations of objects through temporal continuity was possible, at least in theory.

Human psychophysics studies have provided empirical evidence supporting the role of unsupervised temporal contiguity plasticity in visual object recognition. Wallis and Bulthoff found that unsupervised exposure to temporal image sequences of different views of different faces led to performance deficits compared to sequences of the same face (*Wallis and Bülthoff, 2001*; *Wallis and Bülthoff, 1999*). They also pointed out that these results were only observed in similar

face pairs (i.e., low d') rather than very distinct faces (i.e., higher d'). Cox et al. showed that 'swapped' unsupervised experience of pairs of images across saccades could reduce ('break') position tolerance of object discrimination (*Cox et al., 2005*). Balas and Sinha showed that observing object motion can increase both generalization to nearby views and selectivity to exposed views (*Balas and Sinha, 2008*). These behavioral observations revealed that unsupervised temporal contiguity is constantly contributing to the tolerance of object recognition behavior, even in adults, and thus it must be inducing some kind of underlying neural changes somewhere in the brain.

Our human psychophysical results reported here extend this prior work in three ways. First, we measured the learning effects over prolonged periods of time, which allowed us to test for accumulation and saturation. Second, we found that the behavioral learning effect is reversible (*Figure 2C*). Third, we found that this unsupervised learning effect depended on initial task difficulty, which might explain why some studies report stronger effects than others. For example, Wallis and Bulthoff found that the learning effects on view tolerance were only observed in similar face pairs rather than very distinct faces (*Wallis and Bülthoff, 2001*), and those similar face pairs have initial d' that happens to reside in the mid-range where we predict/observe the largest behavioral effects (*Figure 6B*). Third, and most importantly, we designed our unsupervised visual statistical manipulations in the same way as previous monkey neurophysiology experiments, which allowed us to quantitatively compare our human behavioral results with prior monkey neuronal results.

Because IT is, among other ventral stream areas, thought to most directly underlie object discrimination behavior (*DiCarlo et al., 2012*; *Ito et al., 1995*; *Rajalingham and DiCarlo, 2019*) and IT plasticity has been found in many studies (*Baker et al., 2002*; *Logothetis et al., 1995*; *Messinger et al., 2001*), reviewed by *Op de Beeck and Baker, 2010*, it is natural to ask if temporally contiguous unsupervised experience also leads to plastic changes in IT neurons. Miyashita and colleagues showed that neurons in the temporal lobe shape their responses during learning of arbitrarily paired images such that each neuron's response becomes more similar to images that were presented nearby in time (*Miyashita, 1988*; *Miyashita, 1993*; *Naya et al., 2003*; *Sakai and Miyashita, 1991*). Li and DiCarlo directly tested the role of unsupervised visual experience in IT neuronal tolerance by manipulating the identities and properties of objects presented consecutively in time (*Li and DiCarlo, 2008*; *Li and DiCarlo, 2010*). They found that, over ~1.5 hr of unsupervised exposure of 'swapped' temporal statistics, the size and position tolerance of IT neuronal responses were significantly modified, and that these changes were not reward or task dependent (*Li and Dicarlo, 2012*). Qualitatively similar temporal continuity-dependent neuronal plasticity has also been observed in rodents during development (*Matteucci and Zoccolan, 2020*).

Although prior experimental work seemed qualitatively consistent with the overarching theoretical idea, it did not demonstrate that the behavioral learning effects could be explained by the IT neural effects. The results of our study here show that those two effects are quantitatively consistent with each other – the behavioral effects can be largely accounted for by the IT neural effects. While this extends the work of others in the area (see Introduction), some studies have reported null results or have proposed alternative mechanisms. Okamura et al. showed that continuous motion or view presentation is not necessary to form tolerance, rather, enough exposure to different views (even in random sequence) can support view-invariant object recognition (*Okamura et al., 2014*). This evidence suggests an alternative, or additional, mechanism to form tolerant object recognition in addition to temporal continuity. Van Meel and Op de Beeck investigated whether temporal continuity experience can alter size-tolerance representation in human LOC using fMRI and reported no observable effects (*Van Meel and Op de Beeck, 2020*). However, because no behavioral learning effects are reported and fMRI signal has limited spatial and temporal resolution, this null result may not be inconsistent with the results presented here. Looking in rodents, Crijns et al. tested the temporal contiguity hypothesis in adult rat primary visual cortex and found that the tolerance in orientation selectivity across spatial frequency was not affected by temporal continuity manipulation (*Crijns et al., 2019*), which may be caused by a different representation mechanism in lower levels of visual hierarchy. On the other hand, Matteucci and Zoccolan reported that reduced temporal continuity experience in early postnatal life led to a loss of complex cell functional properties in rat V1 (*Matteucci and Zoccolan, 2020*). In sum, the literature is still highly varied and future neurophysiological and behavioral experiments are necessary to test the boundary conditions of temporal contiguity induced effects.

## Future directions

We believe that models similar to the one proposed here will be an important future direction in harmonizing results across spatial scales (neurons to behavior) and across species (rodents to primates to humans), such as the studies outlined above. A second future direction is to extend our current overall model to other modalities, like view invariance or position invariance. This could be done by collecting further psychophysical data, adding proper tuning kernels to the current generative IT model, and using the same IT plasticity rule and decoding model. A third future direction is to extend our current model to other objects beyond those that have been tested in monkeys and humans. This could be achieved through testing new IT population responses to new and old objects and then embedding the new objects in the MDG model of the IT population representation space based on neuronal population response similarity. Alternatively, we can use image-computable deep ANN models that quite accurately predict ventral stream neuronal population responses (*Kubilius et al., 2018*; *Yamins et al., 2014*) and use the 'IT' layer to build a much larger representation space of objects. A fourth future direction is to develop new unsupervised learning algorithms that implement some of the core ideas of temporal contiguity learning, but are scaled to produce high-performing visual systems that are competitive with state-of-the-art neural network systems trained by full supervision. Many computational efforts have touched on this direction (*Agrawal et al., 2015*; *Bahroun and Soltoggio, 2017*; *Goroshin et al., 2014*; *Higgins et al., 2016*; *Kheradpisheh et al., 2016*; *Lotter et al., 2016*; *Srivastava et al., 2015*; *Wang and Gupta, 2015*; *Whitney et al., 2016*), and some are just beginning to make predictions about the responses along the ventral stream (*Zhuang et al., 2021*). A key next step will be to put those full-scale models to experimental test at both the neurophysiological and behavioral levels.

# Materials and methods

## Datasets from prior work

To build a quantitative linking model that predicts unsupervised learning effects in humans from neuronal response in IT, we used three experimental datasets: (1) human data: human psychophysics performance data collected with Amazon Mechanical Turk; (2) IT population data: simultaneous recordings of 168 sites with multi-electrode Utah array recordings implanted in monkey IT (from a previous study; *Majaj et al., 2015*); and (3) IT single-site learning data: multi-unit activity recorded with single electrodes in monkey IT (from a previous study; *Li and DiCarlo, 2010*). All processed data are available at https://github.com/jiaxx/temporal_learning_paper (copy archived at swh:1:rev:bb355bb96286db2148c3abdc8f71b5880f657c5f), *Jia, 2021*.

### IT population dataset

Multi-electrode array (Utah arrays) recordings from two awake macaque monkeys provided 168 multi-unit IT neural sites to 64 objects (5460 high-variation naturalistic images) for modeling use. Image presentation was 100 ms, and each image was repeated between 25 and 50 times. Spike counts were binned in the time window 70–170 ms post stimulus presentation and averaged across repetitions to produce a 5760 by 168 neural response pattern array. The 64 exemplar objects come from eight categories (animals, boats, cars, chairs, faces, fruits, planes, and tables). Images were generated by placing a single exemplar object on a randomly drawn natural scene background at a wide range of positions, sizes, and poses. Images were presented at 8° diameter at the center of gaze to awake fixating animals in a rapid serial visual presentation (RSVP) procedure (horizontal black bars indicate stimulus presentation period). See *Majaj et al., 2015* for details.

### IT plasticity dataset

Physiology data of unsupervised IT plasticity effects measured in multi-unit activities recorded at each single electrode in macaque monkey IT cortex were reanalyzed from *Li and DiCarlo, 2010* (n = 42 multi-unit activity (MUA) sites). We refer to these as 'neural sites.' The FR of sorted units was tested in response to preferred (P) and non-preferred (N) objects, each presented at a range of sizes, and were re-tested after different amounts of unsupervised exposure to evaluate the effect of that exposure on those response measures. See (*Li and DiCarlo, 2010*) for details.

## Image generation

We used the same 3D object models as previous published IT-behavior study (*Majaj et al., 2015*) and applied the same rendering mechanism (ray-tracing software) to each 3D object while parametrically varying its position, rotation, and size, and projected on a randomly chosen unique natural background (out of a pool of 130 images) to generate new test image examples. All images were achromatic. The ground truth of each image was the identity of the generating 3D model, and this was used to evaluate performance accuracy. This naturalistic image generation allows us to gain full control of all the object-related metadata in the images while preserving a relatively natural COR experience.

For each object, we predefine a 'baseline view' (i.e., exact center of gaze, size of ~2° or 1/3 of the diameter of the image, and canonical pose; see Methods of *Majaj et al., 2015*; *Rajalingham et al., 2018*). Variations in size, position, and rotation are transformations relative to baseline view of the object. Since our focus here was unsupervised learning of size-tolerant object selectivity, we intentionally introduced more images that only vary in size to measure size tolerance. Medium-sized objects were the 'baseline' size (~2°). Small-sized objects were 0.5× of baseline (~1°). Big-sized objects were 2× that of baseline (~4°). All test images for different sizes were generated with random naturalistic backgrounds. We thus created a set of 240 'size test' images per object (i.e., 80 images per object at each of the three test sizes). The final test images were each 512 × 512 pixels and were always presented to the subject at a total extent of ~7° of visual angle at the center of gaze (as in the prior neurophysiology studies above).

To neutralize possible size-specific attentional effects and possible size-specific adaptation effects, we presented these 'size test' images intermixed with other 'cover' images of the same objects. These cover images were generated using mild variation in all of the object view parameters. Specifically, we sampled randomly and uniformly from the following ranges: [−1.2°,+1.2°] for object position in both azimuth (h) and elevation (v); [−2.4°,+2.4°] for rotation in all three axes; and [x0.7, x1.3] for size. These cover images were mixed randomly with the 'size test' images (above) at a ratio of 1 cover image per 'size test' image to generate a set of psychophysical test images for each subject (illustrated in *Figure 1—figure supplement 1B*). The behavioral results from the cover images were not part of the analyses.

## Human psychophysics and analysis

All human experiments were done in accordance with the MIT Committee on the Use of Humans as Experimental Subjects (COUHES). We used Amazon Mechanical Turk (MTurk), an online platform where subjects can participate in non-profit psychophysical experiments for payment based on the duration of the task. In the description of each task, it is clearly stated that participation is voluntary and subjects may quit at any time. Subjects can preview each task before agreeing to participate. Subjects will also be informed that anonymity is assured and the researchers will not receive any personal information. MTurk requires subjects to read task descriptions before agreeing to participate. If subjects successfully complete the task, they anonymously receive payment through the MTurk interface. Since it is easier and faster to recruit subjects through MTurk, we can collect a much larger dataset than traditional in-lab human psychophysics.

A total of 505 (174 subjects in *Figure 2* and 331 subjects in *Figure 6*) subjects successfully completed our tasks published through Amazon's Mechanical Turk. Subjects who failed to complete the task or follow the instructions were rejected. Aspects of COR performance were measured based on the behavioral report following each test image presentation (*Rajalingham et al., 2018*). Previous work compared the results of COR tasks measured in the laboratory setting with controlled viewing with results measured via Amazon MTurk and found virtually identical results (Pearson correlation 0.94 ± 0.01; from *Majaj et al., 2015*).

Each behavioral experiment contained two types of phases: a *test phase* in which specific aspects of object discrimination performance were measured (see below) and an *exposure phase* in which pairs of temporally contiguous images were experienced (see *Figure 1*). The main experiment consisted of five test phases (200 trials each; 6–8 min) and four interleaved exposure phases (400 exposure events each; 12–20 min) that together allowed us to measure exposure-induced changes in size-specific object discrimination over time (total experiment time ranged from 90 min to 120 min).

## Test phase

Our goal was to measure the discriminability of targeted (exposed) pairs of objects at targeted (exposed) sizes (and, as references, we also measured discriminability for control object pairs and for target objects at a non-exposed size). Conceptually, each such discrimination sub-task (size-specific object discrimination task) is a generated set of images from object A at a specific size that must be discriminated from a generated set of images of object B at a specific size, and mapped to the same object at a medium size (e.g., see *Figure 1B*, choice images). For clarity, we note that, given this design, the only variation in each of these sub-task image test sets was the image background. These size-specific sub-tasks were randomly interleaved with cover trials to disguise this underlying fact from the subject (see *Figure 1—figure supplement 1B*).

To measure performance on each sub-task, we used a 2AFC design. Each 2AFC trial started with a central fixation point. Subjects were requested to fixate the black fixation point because the test image was always presented briefly at that location and they might miss it otherwise. After 500 ms, the fixation dot disappeared and a test image appeared centered at dot location (center of the screen) for 100 ms, followed by the presentation of two 'choice' images presented on the left and right of the screen (*Figure 1A*). One of the choice images always matched the identity (or category) of the object that was used to generate the test image and was thus the correct choice, and its location was randomly assigned on each trial (50% on the right and 50% on the left). After mouse-clicking a choice image, the subject was given another fixation point (i.e., the next test phase trial began). No feedback on correctness of the choice was given.

To measure size-specific discrimination performance, we created size-specific 2AFC sub-tasks. Specifically, each sub-task was a balanced (i.e., 50%/50%) set of size-specific test images generated from objects A and B (see above), and the two choices presented after each test image were 'clean' examples of objects A and B at a standard ('medium') size (*Figure 1A*).

For each subject, the test images were pseudorandomly drawn from a test image pool that contained the desired number of 'size test' images and cover images (*Figure 1—figure supplement 1B*). Among the 200 trials (50 test images of each test object; four objects in total), 40% contained the 'size test' images (20 for each object; 10 for small and 10 for big), 10% contained baseline views (medium size; five for each object), and the remaining 50% test images were 'cover images' (see above) that were not used in analyses (see *Figure 1—figure supplement 1B* for example test images). The number of test images for target and control object pairs was thus balanced. The number of test images for small and big sizes was also balanced regardless of exposure type. As a result, for each subject, we created six size-specific 2AFC sub-tasks in total (three different sizes for each object pair) regardless of exposure type. The number of test images for target and control face pairs at different sizes in each test phase is specified in *Figure 1—figure supplement 1B*.

To evaluate exposure-induced learning effects, we only calculated the discrimination performance of three exposure-relevant sub-tasks (preplanned, *Figure 1B*): (1) the sub-task with exposed (target) objects at the exposure-manipulated size (*Figure 1B*, red or blue d'); (2) the sub-task with non-exposed (control) objects at the exposure-manipulated size (*Figure 1B*, black d'); and (3) the sub-task with exposed (target) objects at the non-manipulated size (*Figure 1B*, dashed black d'). For example, one subject might have been randomly assigned to exposure type = (experiment u1, swapped condition), target size = (big size), target objects = (face A, face B), control objects = (face C, face D). In this example, each test phase aimed to measure performance (d') on three specific sub-tasks: [face A big vs. face B big], [face C big vs. face D big], and [face A small vs. face B small].

The sizes of the subject groups are provided in Results. The test trials for size and objects were always balanced in each subject group. In *Figure 2*, the subject groups differ in the exposure type (three subject groups). In each of these three groups, the target exposure size was the big size, and within each group, the specific face objects for target and control were randomly selected for each subject. In *Figure 6*, the 13 subject groups correspond to the 13 sub-tasks that were targeted for exposure (see below). Within each subject group, the targeted type of object (i.e., face or basic level) and the targeted exposure size (i.e., medium-big or medium-small) was the same for all subjects, and within each group, the specific objects for target and control were randomly selected within the targeted type.

We computed the d' for each exposure-relevant sub-task (typically three d' values for each subject group; see *Figure 1B*) based on the population (pooled) confusion matrix of the entire subject

group. For each sub-task, we constructed a 2 × 2 confusion matrix by directly filling the behavioral choices into hit, miss, false alarm, and correct rejection according to the stimuli and response of each trial (*Figure 1B*). From the pooled confusion matrix, we computed the d' for each sub-task. We used standard signal detection theory to compute d's from the confusion matrix (d' = Z(TPR) − Z(FPR), where Z is the inverse of the cumulative Gaussian distribution function, and TPR and FPR are true-positive and false-positive rates, respectively). The d' value was bounded within −7.4 to 7.4 (via an epsilon of 0.0001). The mean d' for each sub-task of each subject group was determined by averaging the d' calculated from each bootstrapped subjects sample (which converges to the d' of the pooled confusion matrix). The error bar (bootstrapped standard error) of performance represents the standard deviation of population pooled d' over all bootstrap samples (1000 samples in each case), which is performed by sampling with replacement across all trials (aggregated for each subject group). p-value is directly estimated from the bootstrapped distribution of performance change by comparing to no change condition, which is by definition 0.

## Exposure phase

Each exposure trial (a.k.a. exposure 'event') in the exposure phase was intended to deliver a pair of temporally contiguous images at the center of gaze. Each trial initiated with the presentation of a small, central black dot (~0.5°), and the subject was required to mouse-click on that dot (this is intended to naturally bring the center of gaze to the dot). Immediately after a successful mouse-click (within 0.5° of the dot), two images were presented sequentially at the location of the black dot. Each image was shown for 100 ms with no time lag between them. After the event, the black dot reappeared at a new, randomly chosen location (out of nine possible locations) on the screen (i.e., the next exposure trial began). The details of those images are described below in the context of the specific experiments carried out.

Because we here focused on the effects of unsupervised exposure events on size tolerance, the size of object in each of the two sequential images was always different and always included the medium ('baseline') size: either big-sized objects paired with medium-sized objects or small-sized objects paired with medium-sized objects. In either variant, the order of those two images was counterbalanced, as in *Li and DiCarlo, 2010* (e.g., approximately half of the events transitioned from medium to big objects and the other half from big to medium objects; signified by the double-headed arrows in *Figure 1B*).

## Flavors of unsupervised exposure

Following prior work (*Cox et al., 2005*; *Li and DiCarlo, 2010*; *Wallis and Bülthoff, 2001*), there are two basic flavors of unsupervised exposure. The first flavor is referred to as the swapped exposure, in which the two images within each exposure event are generated from *different* objects (here, at different sizes). Based on prior work (*Cox et al., 2005*; *Wallis and Bülthoff, 2001*), this exposure flavor is expected to gradually 'break' (disrupt) size-tolerance discrimination of those two objects. The second flavor is non-swapped exposure, in which the two images are generated from the *same* object (here, at different sizes). While this have been less studies in human psychophysics, based on prior IT neurophysiology results (*Li and DiCarlo, 2010*), this exposure flavor is expected to gradually build size-tolerant discrimination of those two objects.

## Experimental designs

Our main experimental goal was to test the directions, magnitudes, and temporal profiles of changes In size-tolerant object discrimination (assessed in the test phases, see above) resulting from different types of unsupervised exposure conditions (*Figures 1* and *2*). To do that, we deployed the two flavors (above) in three types of unsupervised experience types (u), and each subject was tested in only one of those three types. The first type (u1) was a series of swapped exposure epochs (intuitively, this aims for maximal 'breaking'). The second type (u2) was a series of non-swapped exposure epochs (intuitively, this aims for maximal 'building'). The third type (u3) was two swapped exposure epochs followed by two non-swapped exposure epochs (intuitively, this aims to test the reversibility of the unsupervised learning).

Each type of experiment lasted for about 90 min, and each consisted of nine phases in total: five test phases (200 test images each) and four exposure epochs (400 exposure events in each epoch;

*Figure 1A*). This experiment was done with face objects only in a total of 174 subjects over all conditions (u1 = 102 subjects, u2 = 36 subjects, u3 = 37 subjects).

Our secondary experimental goal was to study how learning effect depends on the perceptual similarity of the exposed objects. To do this, we chose pairs of objects to cover a wide range of initial discrimination difficulties. Intuitively, it is easier to discriminate an elephant from a pear than it is to discriminate an apple from a pear. Specifically, we chose a total of 13 size-specific object pairs selected from a set of eight face objects (n = 10 pairs) and six basic-level objects (n = 3 pairs). For subjects being exposed to faces, the control objects were also faces; for subjects exposed to basic-level objects, the control objects were other basic-level objects. These 13 pairs were selected based on pilot experiments that suggested that they would span a range of initial discrimination performance. Indeed, when tested in the full experiment (below), we found that mean human initial discrimination difficulties ranged broadly (d' range: 0.4–2.6, based on the first test phase). We thus ran 13 groups of subjects (i.e., one group per target object/size pair) with ~20–40 subjects per group. Because the goal here was to test the *magnitude* of size-specific learning (not the time course), we tested only the 'swapped' flavor of unsupervised experience using just one long exposure epoch (consisting of 800 exposure events). Each subject was exposed with only one pair of objects and was exposed to one size variant of the exposure: small-medium-size swapping or medium-big-size swapping. We bracketed that unsupervised exposure with one pre-exposure test phase (200 trials) and one post-exposure test phase (also 200 trials). The learning effect was always measured at the exposed size (e.g., if exposed with small-medium swapping, the learning effect was measured as the performance change of small-size discrimination task of the exposed object pair), subtracting the performance change for control object pair at the exposed size (all exactly analogous to *Figure 2A*).

When conducting multiple tests of the same null hypothesis and considering any one of those tests to reject that null hypothesis, this results in an increase in the likelihood of incorrectly rejecting the null hypothesis by pure chance. To set an appropriate null rejection level, a correction for multiple comparison (e.g., Bonferroni correction or FDR) needs to be conducted, which corrects the alpha level for each test to account for the number of tests of the same null hypothesis. In our testing of learning effects over exposure amount (*Figure 2*), we are not asking whether the learning effect for any exposure amount is different from 0, which would require multiple-comparison correction for number of tests. Instead, each point is a single test of a different null hypothesis: 'There is no learning effect at exposure amount x.' These results demonstrate how learning effect changes as a function of exposure time for different exposure types. Therefore, we do not believe that the multiple-comparison correction is applicable in this situation. In the statistical test for learning effect of different tasks (*Figure 6*), the dependent variables are the observed learning effects for tasks that differ in initial task difficulty. We are asking whether the learning effect that is measured at 800 exposures for a given task is significantly different from 0 (the null hypothesis for all tasks). Thus, there is only one comparison for each dependent variable; therefore, we believe that a multiple-comparison correction is not necessary here. If we were asking whether there is any learning effect observed for any one of the measured tasks given the exposure, then a multiple-comparison correction would be necessary, but that is not the question being asked here. Instead, we are simply showing the trend of the effect size for each tested task, with bootstrapped standard deviations of the mean, to demonstrate the relationship between initial task difficulty and the learning effect size.

## Generative IT model

We modeled the IT neuronal population response based on the IT population dataset collected from monkey IT cortex with a MDG model. This model assumes that the distribution of IT population response (the distribution the mean responses of individual IT neurons to all images of an object) to each object is Gaussian-like. We tested this hypothesis with a normality test and found that 81.25% (52 out of 64 distributions for 64 objects) of the IT population response distributions were Gaussian (reject when p<0.01). This MDG model preserves the covariance matrix of neuronal responses to all 64 objects that have been tested in monkey IT cortex. A random draw (of a 64 × 1 vector) from the MDG is conceptualized as the average response (over image repetitions) of a simulated IT recording site to each of the 64 objects. To generate the simulated IT tuning over changes in object size, for each simulated IT site, we multiplied (outer product) that 64 × 1 vector with a randomly chosen size-tuning kernel (1 × 3 vector) that was randomly sampled from a batch of size-tuning curves (*Figure 3—figure supplement 1B*) that we had obtained by fitting curves to real IT responses across

changes in presented object size (n = 168 recording sites; data from *Majaj et al., 2015*). This gives rise to perfectly size-tolerant simulated IT neurons (i.e., by construction, the tuning over object identity and over size are perfectly separable). Note that this produced a population of simulated IT neurons with a broad range of size tuning, but with that range approximating that observed across actual IT neurons. The distribution of the variance across size (a.k.a. variance across size reflects the shape of size-tuning curve, e.g., 0 variance corresponds to a flat tuning curve across sizes) is shown in *Figure 3—figure supplement 1C*. To introduce more biological realism and to approximate the fact that each image is presented on a random background, we randomly jittered each value in the $64 \times 3$ matrix by a zero mean, iid shift of each matrix element (randomly drawn from the distribution of variance across image exemplars for each object from IT neural data [*Figure 3—figure supplement 1D*; $\sigma^2_{clutter}$]). Given this procedure, we could generate a potentially infinite number of simulated IT neurons and their (mean) responses to each and every image condition of interest. We verified that, even with the simplifying assumptions imposed here, the population responses of simulated IT populations were quite similar to the actual IT neural population responses (in the sense of image distances in the IT population space [*Figure 3B*] and variance level [*Figure 3D*]).

To generate a hypothetical IT neural (model) population, we simply repeated the above process to obtain the requested number of model neurons in the simulated population (note: the MDG and the size-tuning kernel pool was always fixed). In addition, when we 'recorded' from these neurons (e. g., in *Figure 3A*), we additionally added response 'noise' that was independently drawn on each repetition of the same image ($\sigma^2_{repeats}$; mean zero, variance scaled with the mean to approximate known IT Poisson repetition 'noise'; *Figure 3—figure supplement 1E*).

## IT-to-behavior-linking model

To generate behavioral performance predictions from model IT population responses, we applied a previously defined IT-to-recognition-behavior-linking model (*Majaj et al., 2015*). In that study, the authors used actual IT neural population responses to show that a set of possible IT-to-behavioral-linking models could each accurately describe and predict human performance on all tested recognition tasks within the reliability limits of the data. We here used one of the simplest, most biological plausible of those models – a linking model that seeks to infer the test image's true label by computing the Pearson correlation between the mean IT population response to each possible object class (computed on the IT response to the training images) and the IT population response evoked by the current test image (note that test images are never used in the training of decoders). In other words, the model's 'choice' of object category for each test image was taken to be the choice object whose (simulated) IT population mean (over the training images) was closest to the population vector evoked by the current test image. The only difference from the prior work (*Majaj et al., 2015*) is that here we used simulated IT neurons (see Generative IT model) to drive the 'behavior' of the model. (Note that the linking model has two key hyperparameters [see Results] and, for each simulation run, we held those constant.)

Since the model (IT population + linking model) could now be treated as a behaving 'subject,' we analyzed the behavioral choices in exactly the same way as the actual human behavioral choices to arrive at d' values that could be directly compared (i.e., generate a confusion matrix for each 2AFC sub-task, see above).

Similarly, to test a new model 'subject,' we simply generated an entirely new IT model population (see above) and then found the parameters of the IT-to-behavior-linking model for that subject.

To simulate human lapses (see Results), we introduced a (fixed) percentage of trials in each of the 'behavioral' confusion matrices where model responses were randomly chosen. Note that, when initial d' is below ~2, the lapse rate most consistent with the data (9%) has little influence on measurable performance (see *Figure 6A*) and thus only a minor effect on the model in *Figure 5*. Therefore, all predictions in *Figure 5* were made with 0% lapse rate.

## Unsupervised IT plasticity model

We built a descriptive (non-mechanistic) learning rule with the same fundamental concept as previous computational models of temporal continuity learning (*Földiák, 1990*; *Földiák, 1991*; *Sprekeler et al., 2007*; *Wiskott and Sejnowski, 2002*), except its mathematical implementation. In our setup, there are always only two images in each exposure event (a leading image and a lagging

image). Our plasticity rule states that, after each exposure event, the modification of the mean FRresponse to the leading image is updated as follows:

$$\Delta \mathrm{FR}_{\mathrm{leading}} = \alpha(\mathrm{FR}_{\mathrm{lagging}} - \mathrm{FR}_{\mathrm{leading}})$$

This plasticity rule tends to reduce the response difference between two exposed images (i.e., it tends to create response stability over time, assuming that the statistics of the future are similar). In our overall model, we apply this plasticity rule to each and every simulated IT neuron (true) after each and every exposure event. Note that, under repeated exposure events, the FR to all images will continue to change until there is no difference in responses to the leading and lagging images, which means that the responses will eventually reach a steady state.

Compared with previous plasticity rules (e.g., Hebbian rule) for temporal continuity learning, our plasticity rule is relatively simple. Our plasticity rule updates each IT unit's output FR directly rather than its input weights (*Földiák, 1991*). Based on immediate activity history, our learning rule continuously changes each unit's output by pulling its responses to consecutive images closer until reaching steady state. This learning rule has several features. First, it is temporally asymmetric, which means that the direction of rate change of the leading image depends on the sequence of leading and lagging image. In other words, the response to the lagging image is going to pull the response to the leading image toward it. However, since our experiments randomized the leading and lagging images on each exposure trial, this results in a change in the response to both images rather than an asymmetric change. Second, the effect of our plasticity rule is constrained to exposed image pairs and ignores any correlation in the neural representation space. Even though we do not yet have experimental data to accurately generalize the plasticity rule further than what has been presented in this paper, it is potentially generalizable to other types of tolerance (position, pose) and to other exposure paradigms.

The neural plasticity data were collected by selecting preferred (P) and non-preferred (N) images for each unit, which are two different objects (*Li and DiCarlo, 2010*). We set the neural plasticity rate of the simulated neurons to match that observed in biological IT neurons. To do this, we focused on the same initial high d' regime as the neural plasticity data were collected. Specifically, for each simulated IT site, objects P and N were chosen independently out of the 64 objects based on its mean response to each object (most likely to be in the high d' regime). The plasticity rule was applied to each simulated site as it underwent unsupervised exposure, with the neural response function updated based on its responses to images containing objects P and N at the exposed sizes (see Materials and methods: Generative IT model for details). Because of the initial randomness in the size-tuning kernel selected for each neural site (*Figure 3—figure supplement 1B, C*) and the clutter variance introduced for the site's responses to different image exemplars (*Figure 3—figure supplement 1D*), the response profile of each simulated neuron is unique and thus the updated direction during simulated unsupervised learning is not always in the same direction across the population. The averaged learning effect across all simulated neurons was then computed (as if these neurons had been observed in an experiment) and that simulated learning effect was compared to the averaged learning effect observed in the biological IT neurons. The plasticity rate was optimized to minimize this difference. The (fixed) plasticity rate determined in this way could then be used with the plasticity rule to compute the expected individual IT neuronal response pattern changes to any pair images for which the image-driven responses are both known. In this study, that means we could apply it to any images in the space of the generative model of IT, but we note that this same plasticity rule could be applied to other models of IT responses (e.g., those from contemporary image-computable models; *Kubilius et al., 2018*; *Yamins et al., 2014*). However, it is important to note that the learning rate value is determined by the IT population statistics and the plasticity rule chosen here and thus should not be taken as a universal value. Changes in the statistics of the simulated IT population (i.e., covariance matrix, variance across sizes or clutter variance, etc.) can influence the initial state of the IT population, and as a consequence influence both plasticity rate value and the predicted changes for each simulated IT neuron.

The plasticity rate that best matches neural data is 0.0016 nru per exposure event (nru = normalized response units). The normalized response is calculated by $\Delta(P - N)/(P - N)$, where P and N represent the z-scored FR (across all objects) to preferred and non-preferred objects. Z-score is measured in terms of standard deviations from the mean. Therefore, 1 normalized

response unit is 1 std of the response (FR) distribution across all tested objects. Since the mean multi-unit FR is $90 \pm 23$ spk/s (std across objects) for the IT population across 64 objects, we estimate that 1 nru is ~23 spk/s. Therefore, 0.0016 nru corresponds to a FR change of ~0.035 spk/s per exposure event, which means that ~30 exposure events of this kind would give rise to 1 spk/s change in P vs. N selectivity.

## Acknowledgements

This work was supported in part by a grant from the National Institutes of Health to JJD (2-RO1-EY014970-06) and by the Simons Foundation (SCGB [325500] to JJD). We would like to thank Charles Cadieu, Dan Yamins, Ethan Solomon, Nuo Li, Elias Issa, Rishi Rajalingham, and Arash Afraz for helpful discussions.

## Additional information

### Funding

| Funder | Grant reference number | Author |
| --- | --- | --- |
| National Institutes of Health | 2-RO1-EY014970-06 | James J DiCarlo |
| Simons Foundation | SCGB [325500] | James J DiCarlo |

The funders had no role in study design, data collection and interpretation, or the decision to submit the work for publication.

### Author contributions

Xiaoxuan Jia, Conceptualization, Data curation, Software, Formal analysis, Validation, Investigation, Visualization, Methodology, Writing - original draft, Writing - review and editing; Ha Hong, Data curation, Software, Methodology, Writing - review and editing; James J DiCarlo, Conceptualization, Resources, Supervision, Funding acquisition, Investigation, Visualization, Methodology, Writing - review and editing

### Author ORCIDs

Xiaoxuan Jia https://orcid.org/0000-0001-5484-9331

### Ethics

Human subjects: All human experiments were done in accordance with the MIT Committee on the Use of Humans as Experimental Subjects (COUHES; the protocol number is 0812003043). We used Amazon Mechanical Turk (MTurk), an online platform where subjects can participate in non-profit psychophysical experiments for payment based on the duration of the task. In the description of each task, it is clearly stated that participation is voluntary and subjects may quit at any time. Subjects can preview each task before agreeing to participate. Subjects will also be informed that anonymity is assured and the researchers will not receive any personal information. MTurk requires subjects to read task descriptions before agreeing to participate. If subjects successfully complete the task, they anonymously receive payment through the MTurk interface.

### Decision letter and Author response

Decision letter https://doi.org/10.7554/eLife.60830.sa1
Author response https://doi.org/10.7554/eLife.60830.sa2

## Additional files

### Supplementary files

- Transparent reporting form

## Data availability

All data generated or analyzed during this study are included in the manuscript and supporting files, in the most useful format (https://github.com/jiaxx/temporal_learning_paper (copy archived at https://archive.softwareheritage.org/swh:1:rev:bb355bb96286db2148c3abdc8f71b5880f657c5f)). Datasets from previous studies (IT population dataset [Majaj et al., 2015] and IT plasticity data [Li & DiCarlo, 2010]) are also compiled in the most useful format and saved at the same Github location. Original datasets for previous studies can be obtained by directly contacting the corresponding authors of those studies ([Majaj et al., 2015] and [Li & DiCarlo, 2010]).

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
