## [Decision Letter]

**Acceptance summary:**

This manuscript addresses a major unknown in the literature. Previous empirical studies on the potential role of unsupervised temporal contiguity learning have observed behavioral effects in relatively difficult object discriminations and changes in neural selectivity for very easy object discriminations at the level of single neurons. Linking these two observations is not trivial, and the current manuscript is a very important step towards filling this gap.

**Decision letter after peer review:**

Thank you for submitting your article "Unsupervised changes in core object recognition behavior are predicted by neural plasticity in inferior temporal cortex" for consideration by *eLife*. Your article has been reviewed by 3 peer reviewers, and the evaluation has been overseen by a Reviewing Editor and Michael Frank as the Senior Editor. The following individuals involved in review of your submission have agreed to reveal their identity: Takeo Watanabe (Reviewer #2); Hans Op de Beeck (Reviewer #3).

The reviewers have discussed the reviews with one another and the Reviewing Editor has drafted this decision to help you prepare a revised submission.

Summary:

This paper raised the very interesting and fundamental question of how unsupervised exposure to objects makes the ventral system learn to better recognize objects in a temporally tolerant way. To address this question, an overall unsupervised computational model was built. The model had three core aspects such as a generative model of a baseline monkey IT population, an IT population and behavior linking model, and an IT plasticity rule that predicts response development of temporally associated object images. Sophisticated human psychophysical experiments were also conducted to examine how the model predicts human object recognition performance. The model with empirically-based monkey IT neural parameters remarkably well predicted the human behavioral results. These results suggest that the unsupervised learning processing successfully linked empirical and theoretical learning results based on monkey IT population to human learning of object recognition.

The reviewers agreed that the manuscript would make a welcome addition to a relatively sparse literature – trying to bridge the behavioral and neurophysiological literature. The general approach was deemed elegant, and the psychophysics results for experiments conducted at this scale on AMT to be surprisingly clean (for the extensive subject training it required). These are arguments that the paper has both the novelty and the quality to be a good fit for the journal.

However, there was also a general sense that the story was somewhat oversold – seemingly providing an almost definitive answer to the questions asked, and in particular on the ability to bridge the two levels. The authors did not convince yet the reviewers that this is an appropriate conclusion. First the model fit does appear to be solid enough, in particular in the high-sensitive range in which most of the neural data were obtained. Furthermore, the conditions between the neurophysiological and the psychophysical experiments were not sufficiently matched, and for too many of the differences we do not know to what extent they matter. There were also substantial issues brought up by reviewers on the modeling side with a reviewer even attempting a crude implementation of the model to understand how parameter free it truly is.

Overall, this is a manuscript with great promises but important caveats were found and controls will need to be done.

Essential revisions:

The manuscript can improve substantially in the message it gives about the level of understanding that has been reached with this work.

1. The results confirm that for large image differences there is a smaller behavioral effect of exposure than needed to really say that the missing link has been resolved. Even a model with a relatively large lapse rate does not explain this. There is very little learning for initial d'>>2, despite that the model with lapse rate 9% still expects so. Also, the lapse rate is a handy solution, but not a very convincing argument. This is a problem, because all the neural findings were obtained with large image differences. The authors do not emphasize this enough, but it should almost be with red letters over the manuscript because it totally changes the interpretation of the findings. As we understand the manuscript, α is estimated in Figure 4 using image pairs with a very high d', and then the prediction of learning effect in Figure 6B fails for such image pairs. Isn't this a major problem? Please comment.

2. The authors made the behavioral task artificially difficult by introducing changes to the test paradigm for which it has not been shown how they might affect the learning effect induced by temporal contiguity exposure at the neural level. This might again affect whether the obtained α can be relied upon. Here are some of the differences. First, in the behavioral experiments, the test phase showed the stimuli on a background. This was not done at the neural level -- do the neural effects of temporal contiguity survive adding a background? Second, in the behavioral experiments, cover trials were interleaved to 'disguise' the simplicity of the testing. How would these cover trials impact neural recordings? Third, the behavioral experiment asks not only for invariance across size, but also in position (at least in screen coordinates), and apparently also pose (p. 3). The neural findings are not necessarily generalizable to this more complicated situation. Fourth, the image discriminability is not matched between the new behavioral experiments and the old electrophysiology. Most of the data here are obtained with faces, which is similar to previous behavioral experiments in this domain, but not representative for the very large image differences in electrophysiology. We encourage the authors to think themselves for ways to address these concerns. The ideal way to address these issues would be to re-do the neural data collection. That is obviously out of scope for a revision given that the current manuscript does not present new neural data. New psychophysical data would not help because the changes with respect to neural recordings are necessary to bring performance below ceiling. Maybe this could be done by switching to another dependent variable that remains relevant at ceiling, such as reaction time data? However, this would add an additional layer of complexity and assumptions about how RT relates to neural selectivity. Maybe the authors can consider this. If not, then we would be amenable to the authors (i) acknowledging this discrepancy and toning down their conclusions accordingly, (ii) indicating which differences might be most important (or speculate on why they might not be important, and (iii) mentioning explicitly that further neural recordings are necessary to decide the issue.

3. The authors should also be more complete about the discrepancies in the literature that put further doubt on how accurate our estimate is of the actual effect size and the boundary conditions under which effects of temporal contiguity exposure occur. There are null results in the literature. For example, Crijns et al., 2019 found no effect in rat V1. Nor did a recent human fMRI study (Van Meel et al., 2020, NeuroImage), which used an exposure paradigm with objects and size changes that is very similar to what is used in the current manuscript. This literature is not mentioned, or even cited wrongly (Crijns et al.,). I have a difficulty with bringing all these null results in agreement with the current set of findings. We might overestimate the effect of temporal contiguity based upon how the authors have summarized the literature. In terms of future directions, there is clearly also a need for further neural investigation of the boundary conditions of temporal contiguity effects.

4. There was a level of skepticism regarding how truly parameter free the model actually is. A model that goes from single cell responses to behavior necessarily has quite a few moving parts and associated design choices. For some of these choices it would be comforting to see what happens when they are dropped or altered. To what extent is the apparent zero-parameter match of the learning rates subject to an implicit model fitting by means of design choices? Specifically, the authors use the same size tuning template for all neurons in the population, combined with multiplicative noise. It feels as if this is not a random choice. The reviewers suspect that it will – at least on average – preserve the performance of the classifier across different object scales, because the classifier computes correlation coefficients and hence normalizes responses. It also means that the update during learning for the non-swapped exposure is – again on average – in the same direction for all neurons (either increasing or decreasing the rate, depending on the size tuning template). Is the improvement of discrimination performance in the model independent of the specific size tuning template that is used? Or does it depend on whether the firing rate at the large size is higher or lower than at the medium size? An independent sampling of size tuning for the different neurons would have been the more natural choice. It would be comforting to see the results hold up.

Please note that one of the reviewers actually went through the trouble to actually quickly code up a toy variant of the model. They reported that it appeared as if the learning rate in the model could be controlled by means of setting noise levels, and the consistent size tuning altered how much the model's classification performance improves for the non-swap exposure scenario. It was also noted that the improvement for non-swap and the deterioration for swap is actually even more pronounced without the consistency in size tuning, which maybe speaks in favor of at least the qualitative aspect of the model. The question thus remains whether it is possible to get both learning rates to match the data without the tuning consistency. This is an important point raised and we expect additional comments and controls.

[Editors' note: further revisions were suggested prior to acceptance, as described below.]

Thank you for submitting your article "Unsupervised changes in core object recognition behavior are predicted by neural plasticity in inferior temporal cortex" for consideration by *eLife*. Your article has been reviewed by 2 peer reviewers, and the evaluation has been overseen by a Reviewing Editor and Michael Frank as the Senior Editor. The following individuals involved in review of your submission have agreed to reveal their identity: Takeo Watanabe (Reviewer #2); Hans Op de Beeck (Reviewer #3).

Essential Revisions:

As requested by Reviewer #2, please:

1) Add a brief discussion regarding differences in how objects/faces are encoded in the visual cortex and possible implications (or lack thereof) for the interpretation of results.

2) Confirm that corrections for multiple comparisons were implemented.

[Editors' note: further revisions were suggested prior to acceptance, as described below.]

Thank you for resubmitting your work entitled "Unsupervised changes in core object recognition behavior are predicted by neural plasticity in inferior temporal cortex" for further consideration by *eLife*. Your revised article has been evaluated by Michael Frank (Senior Editor) and a Reviewing Editor.

The manuscript has been improved but there are some remaining issues that need to be addressed, as outlined below:

The argument that you are making regarding the absence of correction for multiple comparisons sounds reasonable, but it is open to debate. While we are sympathetic to your point about letting "the reader [to] decide on what inferences to draw from the data, confidence intervals and associated statistical test", we also feel that in the current manuscript form, the reader might not realize the absence of corrections. As a compromise, we suggest that you add a paragraph in the methods including your justification to make this clear.

---

## [Author Response]

Summary:This paper raised the very interesting and fundamental question of how unsupervised exposure to objects makes the ventral system learn to better recognize objects in a temporally tolerant way. To address this question, an overall unsupervised computational model was built. The model had three core aspects such as a generative model of a baseline monkey IT population, an IT population and behavior linking model, and an IT plasticity rule that predicts response development of temporally associated object images. Sophisticated human psychophysical experiments were also conducted to examine how the model predicts human object recognition performance. The model with empirically-based monkey IT neural parameters remarkably well predicted the human behavioral results. These results suggest that the unsupervised learning processing successfully linked empirical and theoretical learning results based on monkey IT population to human learning of object recognition.The reviewers agreed that the manuscript would make a welcome addition to a relatively sparse literature – trying to bridge the behavioral and neurophysiological literature. The general approach was deemed elegant, and the psychophysics results for experiments conducted at this scale on AMT to be surprisingly clean (for the extensive subject training it required). These are arguments that the paper has both the novelty and the quality to be a good fit for the journal.

We are pleased to hear that the reviewers appreciate the bridge we aim to build in this line of research and the specific contributions of this particular study. We note that the “clean” Amazon Mechanical Turk (AMT) data is likely a result of the very large number of subjects that the platform allowed us to study.

However, there was also a general sense that the story was somewhat oversold – seemingly providing an almost definitive answer to the questions asked, and in particular on the ability to bridge the two levels. The authors did not convince yet the reviewers that this is an appropriate conclusion. First the model fit does appear to be solid enough, in particular in the high-sensitive range in which most of the neural data were obtained. Furthermore, the conditions between the neurophysiological and the psychophysical experiments were not sufficiently matched, and for too many of the differences we do not know to what extent they matter. There were also substantial issues brought up by reviewers on the modeling side with a reviewer even attempting a crude implementation of the model to understand how parameter free it truly is.Overall, this is a manuscript with great promises but important caveats were found and controls will need to be done.

Our mistake – we did not intend to claim that this study provides a definitive answer to the motivating question. Instead, our goal was to try to clearly explain what we did (which is a complicated set of things to carry out and clearly walk the reader through in a single paper) and then describe how things turned out (a surprisingly good match of the predictions and the data, even though that did not need to be the case). However, on re-reading the paper with these reviewer comments in mind, we agree that our language and phrasing overstates the inferences that can be drawn from these results. As such, we have done our best to modify the text in the abstract, introduction, and discussion to focus on the approach and results, but to avoid language that implies conclusions beyond those results.

In particular, we agree that, because the study we carried out did not show clear psychophysical effects at large initial d’ values (where most of the neural plasticity data were collected), it is impossible for us to know for sure from this study alone that the linkage we infer will indeed hold in that (high initial d’) regime, but only that all indications we have thus far suggest that it might. As such, we now emphasize our uncertainty about what might be going on for high initial d’ tasks, we situate our results in that context, and we proposed potential new future experiments/models to resolve the issue.

In addition, we have made the following changes:

– We carried out additional analysis to test the hypothesis of lapse rate in our human subject pool and justified its value in our overall model (reviewers’ comment 1).

– We added discussion on condition differences between the neurophysiological and the psychophysical experiments. We explain why we see those differences as a strength rather than a weakness of the study, but we also take care in describing the limits of the inferences that we draw from these results, including the possibility that all of the condition differences have fortuitously conspired to result in largely accurate model predictions (reviewers’ comments 1 and 2).

– We added discussion acknowledging the range of results (especially null results) in the literature on the learning effects induced with temporal continuity and we outline future experiments to test potential limits of learning induced via temporal continuity experience (reviewers’ comment 3).

– We clarified that our size tuning curve for each simulated unit is randomly chosen rather than the same for all units (reviewers’ comment 4).

– We do not want to claim our model is parameter free, only that no parameters were tuned in making the predictions about the size of the behavioral effects in the low and mid d’ regimes. We have now clarified that our model is constrained by IT data (by adding a supplemental figure illustrating the variance distributions from IT data) and emphasized it is not fit to the to-be-predicted behavioral changes (reviewers’ comment 4).

Each of these changes is summarized in detail below in response to specific questions.

Essential revisions:The manuscript can improve substantially in the message it gives about the level of understanding that has been reached with this work.1. The results confirm that for large image differences there is a smaller behavioral effect of exposure than needed to really say that the missing link has been resolved. Even a model with a relatively large lapse rate does not explain this. There is very little learning for initial d'>>2, despite that the model with lapse rate 9% still expects so. Also, the lapse rate is a handy solution, but not a very convincing argument. This is a problem, because all the neural findings were obtained with large image differences. The authors do not emphasize this enough, but it should almost be with red letters over the manuscript because it totally changes the interpretation of the findings. As we understand the manuscript, α is estimated in Figure 4 using image pairs with a very high d', and then the prediction of learning effect in Figure 6B fails for such image pairs. Isn't this a major problem? Please comment.

We agree that we should have explained this issue far earlier in the paper, and we have now done so. To summarize:

The reviewers are correct that most of the IT neural data were collected for relatively large object differences (e.g. dog vs. boat). With those large differences, the typical human behavioral performance at discriminating among objects is likely large (e.g. d’ greater than ~3; though we do not have tests with the actual objects used in the neural study as those images were not derived from 3D object models). If human subjects have behavioral lapses (i.e. they make errors un-related to the sensory evidence provided), then our modeling suggests that there is a ceiling of measurable performance and thus the ability to detect behavioral changes of the magnitudes that the neural data predict in this behavioral performance regime (a.k.a. easy discrimination tasks) would be essentially minimum or impossible given the exposure dose we provided here (see Figure 6A). In fact, attentional lapse is almost inevitable in most psychophysics tests (Prins, 2012). As such, we do not consider this to be a “handy solution” (to use the Reviewer’s phrase), but an experimental limitation that is predicted by standard psychophysical models (i.e. a non-zero amount of stochasticity caused by attentional state fluctuation or mistakes in motor action) (Prins, 2012; Wichmann and Hill, 2001).

Because we were aware of this experimental limitation, at the start of our study we focused on object discrimination tasks with initially moderate d’ values (i.e. moderately difficult tasks). The key idea that we employed was that we could tune the plasticity of the simulated IT population in the same d’ regime where the neural plasticity data were collected (high initial d’; (Li and DiCarlo, 2010)) by selecting the preferred and non-preferred objects from a total of 64 objects for each recording site (clarified in Methods lines 1347-1354), and then – because we also have a separate model of the initial IT population patterns across all level of image discriminability -- ask how well the overall model predicts in the regime where the behavioral effects can be measured (low to moderate initial d’s). And we report that this prediction is quite quantitatively accurate (but not perfect) in these behavioral performance regimes for a range of exposure types. These results are the main contribution of the study (Figure 5. A, D and E), and we hope that the reviewers will agree that this quantitative correspondence did not need to obtain from the experiments we conducted, and is thus a contribution. Moreover, in some sense, this quantitative correspondence is, at least to us, even more surprising because, as the reviewers correctly points out, we tuned the learning rate of the neural plasticity model based on measurements of neural responses to images in the high behavioral performance regime! This quantitative correspondence is also notable because there are differences in the conditions that were used in the behavioral and (two) neural experiments that the model had to successful bridge (see response to reviewers’ comment 2).

The accuracy of these predictions in the low and moderate initial d’ regime implies that the linkage is likely to apply across all behavioral regimes. However, we agree with the reviewer(s) that we have not directly shown that the linkage is accurate in the high initial d’ regime. Indeed, we did not observe strong behavioral changes in that regime with the unsupervised exposure dose we were able to deliver in this study. Why not? In the paper, we put forward the hypothesis that even small lapse rates (i.e. small levels of stochasticity caused by attentional state fluctuation or mistakes in motor action) prevent us (or anyone) from measuring changes in behavioral accuracy in that regime given the dose of unsupervised exposure we deployed. In the new version, we provide further evidence in support of that hypothesis by showing ceiling performance in subjects measured with basic level tasks (ceiling is not 100%, which indicates lapse rate is not 0; new Supp. Figure 3). However, we agree that an alternative explanation – one that we previously failed to emphasize – is the possibility that no behavioral effects will ever be found in this high initial performance regime, no matter how strong the unsupervised exposure dose (Introduction lines 107-109; Discussion lines 705-707), and that our results presented in this study cannot directly test the linkage of the neural effects to the behavioral effects in this regime (because we found no behavioral effects to predict!). As a result, we cannot yet guarantee that the proposed linkage model holds over all initial d’ regimes.

To further explore and clearly demarcate what we can say and cannot say as a result of our experiments (explained above), we have done the following:

1. To infer the empirical lapse rate of our subject pool, we have estimated the performance ceiling of subjects in our study. To do so, we used half of the trials to rank order the difficulty of all the tasks we tested in Figure 6 from difficult to easy, and then we plotted the mean population performance of the other half of the trials on those rank ordered tasks (panel A). This plot shows a plateau of mean performance around 90% correct, which we take as an estimate of the mean measurable performance ceiling that can be observed in our subject pool even if perfect sensory evidence was provided. From this performance ceiling, we can infer a mean lapse rate (i.e. 20% lapse rate give a 90% performance ceiling) of our subject pool. This is still within a reasonable range in human lapse rate (<20%; (Manning et al., 2018)). Furthermore, the cumulative performance distribution of all bootstrapped population performance from our basic level tasks (panel B) showed a maximum performance at 95%. Combined together, this evidence indicate the existence of a performance ceiling and justify our use of lapse rate in our full model when making predictions of behavioral changes. We have now included this plot as supplemental Figure 3 and added text in results (Results: lines 594-607). We are aware that even though we were able to indirectly estimate the lapse rate of our subject pool, this is not a direct measurement in a well-controlled condition, and we specifically discussed this point in discussion (lines 687-707) and proposed alternative possibilities and future directions.

While we were redoing the old and new model simulations for Figure 6, we found an error in our previous implementation of the lapse rate. That implementation error resulted in the previously submitted Figure 6B plot having the correct qualitative form, but not quantitatively correct values. We have now double checked the simulations in Figure 6 to fix this problem. And in the resubmission, we have updated Figure 6 to show the predicted learning effect with three different lapse rates to help make clear how lapse rate affects both the observed initial performance (x axis) and the learning effect size that is expected to be observed (y axis). Specifically, we now include the base model with no lapse rate at all (i.e. lapse rate = 0%), the base model with a lapse rate determined from the maximum task performance observed in our subject pool (accuracy ~95.5%, lapse rate = 9%), and the lapse rate inferred from the sigmoidal estimate of the performance ceiling in our subject pool (accuracy ~90%, lapse rate = 20%; see Supp. Figure 3).

2. We now explain in the Introduction that we have little to no sensitivity to measure behavioral effects in the high initial d’ regime where the original neural data were collected, so we needed a different approach (Introduction: lines 90-100).

3. We now emphasize that model predictions are made by tuning the unsupervised learning portion of the model to neural data in the high initial d’ regime (where the neural data were collected) and then making full model predictions in the low to moderate initial d’ regimes where behavioral learning effects are predicted to occur (Introduction: lines 90-100; Discussion: lines 664-673; Methods: lines 1347-1354).

4. We modified the writing in the introduction and in the discussion to make it clear that – because of the limits of the psychophysical experiments that we did -- we (the field) do not yet know what is going on in the high initial d’ regime (where the original neural data were collected), and, based on our modeling, we propose future psychophysical experiments that should produce effects in that regime (if the overall model is correct). (Introduction: lines 108-110; Discussion: 703-705)

Specific changes in manuscript text are (all modified text in the resubmitted manuscript are highlighted for visibility):

To address point 1 above, we modified result section for Figure 6 (lines 578-614):

“We found that our overall model quite naturally – for the reasons outlined above – predicted the smaller learning effect for initially difficult tasks (the left side of Figure 6B). […] That being said, it also means that the current study is simply not able to test the IT-plasticity to behavioral-learning linkage in the initial high d’ range, and we take that as a target for future experimental work (see Discussion).”

To address point 1 above, we modified the Discussion section (lines 687-707)

“The dependency of behavioral learning effects on initial task difficulty is also an important observation. […] However, our results here cannot rule out the possibility that no behavioral effects will be found in this high initial performance regime, no matter how strong the unsupervised exposure dose. Such a finding would falsify the model presented here.”

To bring up this issue early, in Introduction section (lines 90-109):

“Because human make sensory independent mistakes due to inattentional state, these sensory independent random choices (referred to as lapse rate) set a ceiling in the measurable human behavioral performance (Prins, 2012; Wichmann and Hill, 2001). When tasks are in the saturated regime, it is hard to detect any learning effect as any changes in sensory representation would be hidden by the behavioral ceiling (see later). Therefore, we focused our psychophysical study in the mid-range of task difficulty where learning effects can be measured. However, this meant that the task difficulty in human psychophysics could not be in the basic object regime where the neural data were collected. Thus, to make behavioral predictions from the neural data, we took advantage of the overall model to build this bridge: we first tuned the unsupervised plasticity rule by neural data with basic level object images (Li and DiCarlo, 2010); we then used a generative IT model – capable of simulating the response of each artificial IT neuron for a wide range of image discriminability levels – to make quantitative predictions of behavioral change in the regime where the human behavioral learning effects can be readily measured.

Indeed, our behavioral tests revealed a strong dependency of learning effect on the initial task difficulty, with initially hard (d’<0.5) and initially easy (d’>2.5) COR tasks showing smaller measured learning effects than COR tasks of intermediate initial difficulty. We found that our overall model was quite accurate in its predictions of the direction, magnitude, and time course of the changes in measured human size tolerance in the regime where behavioral effects were readily measured for all of the tested unsupervised experience manipulations. The overall model also predicted how the behavioral effect size depended on the initial d’ once we assume behavioral lapses (Prins, 2012) in the model at approximately the same level as those inferred in our subject pool. We note that, because of the (expected) inability to observed behavioral changes for tasks with initial high d’, this study cannot confirm or refute the hypothesized linkage between IT neural effects and behavioral effects in that particular regime.”

To clearly explain how we determined the learning rate in high d’ regime and applied it to other task difficulties, in the Discussion section (lines 664-673):

“We think it is quite interesting and demonstrative of this approach that the overall model could predict behavioral learning effects in the low initial d’ and moderate initial d’ regime for different exposure types (build or break), even though the images used in those tasks are in a different d’ regime from the specific images used in the neuronal plasticity experiments used to tune the model learning (i.e. those neural experiments were done with images in a high d’ regime). We note that the ability to make such tests in these other d’ regimes is the key reason that we built the overall composite model in the first place: a generative IT model (simulating IT population patterns to a wide range of image conditions) + an IT neural-to-behavior linking model + an IT plasticity model (independent of specific images). End-to-end models such as this allow us to predict effects under conditions that no experiment has yet tested, and to then test some of them as we have done here. That is, the integrated model imparts new generalizable knowledge and understanding.”

In the methods section, we clearly explained how we obtained learning rate and factors that could influence its value (lines 1347-1369):

“The neural plasticity data were collected by selecting preferred (P) and nonpreferred (N) images for each unit, which are two different objects (Li and DiCarlo, 2010). […] Changes in the statistics of the simulated IT population (i.e. covariance matrix, variance across sizes or clutter variance, etc) can influence the initial state of the IT population, and as a consequence influence both plasticity rate value and the predicted changes for each simulated IT neuron.”

To clearly state the limitation of our work in the high d’ regime (point 4 above), we added in introduction (lines 107-109):

“We note that, because of the (expected) inability to observed behavioral changes for tasks with initial high d’, this study cannot confirm or refute the hypothesized linkage between IT neural effects and behavioral effects in that particular regime.”

In the Discussion section (lines 705-707):

“However, our results here cannot rule out the possibility that no behavioral effects will be found in this high initial performance regime, no matter how strong the unsupervised exposure dose. Such a finding would falsify the model presented here.”

2. The authors made the behavioral task artificially difficult by introducing changes to the test paradigm for which it has not been shown how they might affect the learning effect induced by temporal contiguity exposure at the neural level. This might again affect whether the obtained α can be relied upon. Here are some of the differences.

The reviewers are correct in that we did not expose or test the subjects to any of the exact stimuli that were used in the monkey neural experiments. Instead, we rely on a model of IT neural activity that applies over a range of conditions and make behavioral change predictions in image exposure and test conditions where the neural plasticity data were not measured. Because the predictions turn out to be accurate across this range of conditions, we consider this to be a strength, rather than a weakness of the study. If the predictions had turned out to be inaccurate, then each of the differences between the neural and the behavioral data become important to consider in trying to understand where the model is broken (e.g. IT does not actually support this behavior, or it does not support is as modeled, or it is not plastic in humans, etc. etc.). Indeed, if the results had revealed that prediction failure, we would have started carrying out new experiments to try to pin down the condition difference(s) that might isolate the source of the prediction failure. However, that is not what we found.

Of course, there remains the possibility that the match of the model predictions to the empirical data (in the low and mid d’ range) derives from the fortuitous (or unfortunate, depending on your point of view) coincidence of separate effects of each of the differences that the reviewer points out below. In the new manuscript, we have now clearly indicated these differences (Discussion: lines 708-728), and the possibility, however unlikely, that a fortuitous coincidence of model assumption failures is still a possible explanation of our finding of a good alignment between the model predictions and the empirical behavioral data.

For clarity, we describe those model assumptions below as they relate to the condition differences that the reviewers raise.

First, in the behavioral experiments, the test phase showed the stimuli on a background. This was not done at the neural level – do the neural effects of temporal contiguity survive adding a background? Second, in the behavioral experiments, cover trials were interleaved to 'disguise' the simplicity of the testing. How would these cover trials impact neural recordings?

Because both of these questions are asking about condition differences in test images, we addressed them together as follows.

First, it is important to clarify that the main learning effect of temporal continuity is introduced during the exposure phase, where the exposed images are presented without background in both the neural and behavioral experiments. We consider that if neural activity plasticity is induced mainly during unsupervised exposure (as found in the original Li et al. studies), then alterations to the test images relative to the Li et al. studies (e.g. adding background or introducing cover trials during test) should not change the neural plasticity (Discussion: lines 716-718).

Second, it is important to emphasize that the model of (initial) generative IT population activity was derived from IT population recordings with images of objects on naturalistic backgrounds (similar to those used in the test images here). That generative IT model was constrained by those IT data. The IT-COR-behavior linking model that linearly reads out from the model IT population representation to make predictions about human performance behavior is also based on the same IT population recording dataset. Therefore, we here used the same method to generate test images with backgrounds and variations for human psychophysics, so that we can use the IT-COR-behavior model to establish the link between simulated IT population and measured human behavior. Thus, for our modeling purpose and behavioral experiments we aimed to explain, we think the choice of test images is justified. We have added additional text in Discussion (lines 712-716) to clarify this for the reader.

Third, the behavioral experiment asks not only for invariance across size, but also in position (at least in screen coordinates), and apparently also pose (p. 3). The neural findings are not necessarily generalizable to this more complicated situation.

From this comment, it appears that we failed to clearly explain the experiments we did. During the exposure phase, the only variation we introduced is object size (Introduction: lines 83-86). We did not (in this study) consider unsupervised exposures to changes in position or pose. However, during the test phase, we introduced changes in size, position, pose and background. We did this to make sure the subjects were each engaged in each task as a genuine object discrimination task (rather than a pairwise image discrimination task). Thus, most of the test trials were to insure that the subjects were working in this mode, and we the later sorted out the specific image pairs where the prior work allowed us to make predictions (i.e. changes in object size). Thus, most of the test trials were cover trials in that sense. We have now clarified this in section (Discussion: lines 712-716).

In addition, we agree with the reviewer in that all of our results only apply to learning about changes in object size, and we have now further clarified that in the Introduction section (lines 83-85) and we have text in the Discussion section (lines: 817-819) to emphasize that the effects of other types of invariance are still to be tested. We also now state that the model we provide here can be used to make predicting about those behavioral effects, but we cannot yet say if those predictions will be accurate.

Fourth, the image discriminability is not matched between the new behavioral experiments and the old electrophysiology. Most of the data here are obtained with faces, which is similar to previous behavioral experiments in this domain, but not representative for the very large image differences in electrophysiology.

This is addressed in the replies above to reviewers’ comment 1. In short, the model allows us to make predictions across those differences and the predictions turned out to be accurate (lines 664-673).

We encourage the authors to think themselves for ways to address these concerns. The ideal way to address these issues would be to re-do the neural data collection. That is obviously out of scope for a revision given that the current manuscript does not present new neural data. New psychophysical data would not help because the changes with respect to neural recordings are necessary to bring performance below ceiling. Maybe this could be done by switching to another dependent variable that remains relevant at ceiling, such as reaction time data? However, this would add an additional layer of complexity and assumptions about how RT relates to neural selectivity. Maybe the authors can consider this. If not, then we would be amenable to the authors (i) acknowledging this discrepancy and toning down their conclusions accordingly, (ii) indicating which differences might be most important (or speculate on why they might not be important, and (iii) mentioning explicitly that further neural recordings are necessary to decide the issue.

The reviewer has very nicely recapitulated our thought process on how to proceed with changes to the manuscript. As suggested, we have now more narrowly scoped the inferences we can draw on and we have now acknowledged these issues up front in the manuscript (lines 86-87) and extensively discussed in Discussion.

The only future oriented point we differ slightly with the reviewers on this is that, while we agree that more neural recordings might be useful in the low to moderate initial d’ performance regime, to our mind the most important thing that needs to be done next is even more stringent psychophysical studies in the high initial d’ performance regime (or with higher unsupervised exposure dose). Based on the results presented here, the neural data clearly predict that internal accumulated sensory evidence in this regime is strongly weakened by unsupervised exposure, but the linking model – including its lapse component – tells us that those effects will be very difficult to measure. Now that we have the results in the current study, we are on much more solid ground to proceed with those experiments in a follow up study, and if we (or others) fail to find behavioral effects after dealing with the psychophysical challenges, this would importantly falsify the current leading linking model of IT to core object recognition behavior. We discuss this now in the modified Discussion, along with proposals for further neural recordings to further bridge differences between neural and behavioral studies.

Specific changes in manuscript text are in the Discussion section (lines 704-724):

“In this work, since we want to establish a quantitative link between single IT neuronal plasticity and human unsupervised learning effect, we matched many of the conditions to those used in prior IT plasticity work. […] Future neural recordings in non-human primates with behavioral learning measurement could be helpful to directly validate the link suggested by our model using the exact same objects and test images as the ones in our human psychophysics.”

3. The authors should also be more complete about the discrepancies in the literature that put further doubt on how accurate our estimate is of the actual effect size and the boundary conditions under which effects of temporal contiguity exposure occur. There are null results in the literature. For example, Crijns et al., 2019 found no effect in rat V1. Nor did a recent human fMRI study (Van Meel et al., 2020, NeuroImage), which used an exposure paradigm with objects and size changes that is very similar to what is used in the current manuscript. This literature is not mentioned, or even cited wrongly (Crijns et al.,). I have a difficulty with bringing all these null results in agreement with the current set of findings. We might overestimate the effect of temporal contiguity based upon how the authors have summarized the literature. In terms of future directions, there is clearly also a need for further neural investigation of the boundary conditions of temporal contiguity effects.

We have now corrected the citation (line 793-794) and added a paragraph in Discussion to address the null results in the literature and the necessity to test the limits of temporal continuity learning effects (Discussion: lines 795-813).

“Although prior experimental work seemed qualitatively consistent with the overarching theoretical idea, it did not demonstrate that the behavioral learning effects could be explained by the IT neural effects. […] In sum, the literature is still highly varied and future neurophysiological and behavioral experiments are necessary to test the boundary conditions of temporal contiguity induced effects. ”

4. There was a level of skepticism regarding how truly parameter free the model actually is. A model that goes from single cell responses to behavior necessarily has quite a few moving parts and associated design choices.

As the reviewers pointed out, it is impossible to have a parameter free model to go from single neuron plasticity to behavior. Instead, our model is a physiological data constrained model. Here are our design choices. First, we constrained the simulated IT population based on real IT recording data, which includes the IT representation for different objects (simulated by multi-variant Gaussian based on the covariance matrix of the population), variance across stimulus size (simulated by extracting size tuning kernels from real IT neurons), variance across random backgrounds (simulated by introducing variance for image exemplars from the data distribution; Figure 3—figure supplement 1D) and variance across repeats (simulated by introducing variance across repeats with values from the real distribution; Figure 3—figure supplement 1E). We have now added a new supplemental figure to show the distributions of variance (Figure 3—figure supplement 1). These distributions constrained the representation and variance of the simulated IT population, which as a result will influence the initial response profiles of individual simulated neurons.

Second, we constrained the learning rate based on the neuronal plasticity of single IT recording sites. We hypothesized a learning rule with one free parameter – the learning rate, which describes the rate of neural activity change for consecutive images. We tuned the value of that learning rate ONLY by matching the population learning effect of the simulated IT population with the measured learning effect of real IT neurons. Therefore, the learning rate is specific to the statistics of the simulated IT population and is constrained by real IT plasticity data.

Third, we constrained the IT-COR-behavior linking model parameters (specifically, number of training sample and population size) based on human initial performance (In effect, implementing the linking model proposed by prior work (Majaj et al., 2015)).

Most importantly, all of the above parameters were locked before we made predictions of human learning effects (except for a final introduction of lapse rate to explain the ceiling effect for easy tasks). Therefore, our model is not parameter free overall. However, all free parameters were constrained by real data but were not tuned to the behavioral learning effects that we measured here. We emphasize that there is no parameter tuning when predicting the human learning effects. In practice, we did not include the lapse rate in the first model prediction because it only becomes obvious in retrospect and, given the reviewers’ suggestions (above), we have done more work to independently justify the choice of this parameter (Figure 6—figure supplement 1: indication of ceiling performance). In sum, all of the model behavioral effects in Figures 5A, D and E were true (i.e. parameter free) predictions. Prediction in Figure 6B is largely parameter free (new Figure 6B, black dots) with the exception of lapse rate which is reasonably constrained by human ceiling performance (See response to reviewers’ comment 1).

We now emphasize the importance of constraining models by experimental data and made the following text changes:

In the legend of Figure 1 (line 175):

“This study asks if that idea is quantitatively consistent across neural and behavioral data with biological data constrained models.”

In Results Lines 292-298

“In sum, this model can generate a new, statistically typical pattern of IT response over a population of any desired number of simulated IT neural sites to different image exemplars within the representation space of 64 base objects at a range of sizes (here targeting “small”, “medium”, and “big” sizes to be consistent with human behavioral tasks; see Methods for details). The simulated IT population responses were all constrained by recorded IT population statistics (Figure 3—figure supplement 1). These statistics define the initial simulated IT population response patterns and thus they ultimately influence the predicted unsupervised neural plasticity effects and the predicted behavioral consequences of those neural effects.”

Lines 458-462

“At this point, we could – without any further parameter tuning – combine each of these three model components into a single overall model that predicts the direction, magnitude and time course of human unsupervised learning effects that should result from any unsupervised learning experiment using this exposure paradigm (pairwise temporal image statistics).”

Lines 474-477:

“Again we emphasize that, while the overall model relies heavily on data and parameters derived explicitly or implicitly from these prior studies (Li and DiCarlo, 2010; Majaj et al., 2015), no components or parameters of this model and nor its predictions depended on the behavioral data collected in this study.”

Lines 531-535:

“In sum, it is not the case that any model of the form we have built here will produce the correct predictions – proper (biological) setting of the unsupervised IT plasticity rate and proper (biological) setting of the IT-to-COR-behavior linkage model are both critical. It is important to note that we did not tune these biologically constrained hyperparameters based on fitting the unsupervised behavioral learning effects in Figure 2 – they were derived in accordance with prior neurobiological work as outlined above.”

In Discussion (lines 637-639):

“Each of these three model components was guided by prior work and constrained by recorded IT data to set its parameters, and the combined overall computational model was directly used to predict human learning effect without further parameter tuning.”

For some of these choices it would be comforting to see what happens when they are dropped or altered.

We strongly resonate with this comment and it was our first reaction to the match between the model and the results. That is, we wondered, maybe all models produce the same result? We took exactly the strategy the reviewer proposed – we tested how altered parameters (i.e. changing the model) would influence the match to the results in Figure 5. Here, we altered 3 parameters: plasticity rate, number of training samples and number of recording sites (lines 506-535). We found that the plasticity rate can significantly alter the predicted learning effect magnitude so that it no longer matches the observed learning effect magnitude. We also found there is a manifold in the decoder parameter space that matches human initial performance. As long as the parameters are on this manifold, the human learning effect predicted by each of those models is similar to the measured learning effect. These counterfactual results gave us comfort that the observed match between the (original) model and the behavioral data was interesting and hence we included that in the original submission.

Modified text in Results (lines 526-525):

“In contrast, when we built model variants in which the choices of the two hyperparameters did not match human initial performance, the unsupervised learning effect predicted by the overall model clearly differed from the observed human learning effect. Specifically, when an overall model starts off with “super-human” performance, it overpredicted the learning effect, and when a different model starts off as “sub-human” it underpredicted the learning effect.

In sum, it is not the case that any model of the form we have built here will produce the correct predictions – proper (biological) setting of the unsupervised IT plasticity rate and proper (biological) setting of the IT-to-COR-behavior linkage model are both critical. It is important to note that we did not tune these biologically constrained hyperparameters based on fitting the unsupervised behavioral learning effects in Figure 2 – they were derived in accordance with prior neurobiological work as outlined above.”

It is worth noting that the simulated IT population responses were all constrained by recorded IT population statistics. Changes in the statistics of the simulated IT population (i.e. covariance matrix, size tuning or cluster variance, etc) can influence the initial state of the IT population, and as a consequence influence both learning rate value and the prediction. However, since this deviation from real IT population lives in a much higher dimensional parameter space, we did not explore further in this direction, but rather, locked the generative IT model according to real IT population statistics. Future work with proper constrains could be done in this direction.

Modified text in Results (lines 295-298):

“The simulated IT population responses were all constrained by recorded IT population statistics (Figure 3—figure supplement 1). These statistics define the initial simulated IT population response patterns and thus they ultimately influence the predicted unsupervised neural plasticity effects and the predicted behavioral consequences of those neural effects.”

Modified text in Methods (lines 1356-1360):

“However, it is important to note that the learning rate value is determined by the IT population statistics and the plasticity rule chosen here and thus should not be taken as a universal value. Changes in the statistics of the simulated IT population (i.e. covariance matrix, variance across sizes or clutter variance, etc) can influence the initial state of the IT population, and as a consequence influence both plasticity rate value and the predicted changes for each simulated IT neuron.”

To what extent is the apparent zero-parameter match of the learning rates subject to an implicit model fitting by means of design choices? Specifically, the authors use the same size tuning template for all neurons in the population, combined with multiplicative noise. It feels as if this is not a random choice. The reviewers suspect that it will – at least on average – preserve the performance of the classifier across different object scales, because the classifier computes correlation coefficients and hence normalizes responses. It also means that the update during learning for the non-swapped exposure is – again on average – in the same direction for all neurons (either increasing or decreasing the rate, depending on the size tuning template).

First, we would like to clarify that we did not (and do not) claim a zero-parameter match for estimating the neural plasticity rate. Instead, the neural plasticity rate is tuned so that the simulated IT neurons match the amplitude of changes observed in the prior IT plasticity studies (Li and DiCarlo, 2010). Because that plasticity rate is tuned using simulated neurons from the generative model of IT and the size tuning kernels, any changes in those response profiles might change the value of plasticity rate needed to match those prior data. Note that, because of this, we do not view the plasticity rate that we report as a key claim of our study.

Second, this comment implies that we were not clear about our method of simulating neural size tuning in our original manuscript. To clarify, for each simulated IT site, we randomly draw a size tuning kernel from a pool of ~168 size tuning kernels (fit to the mean size tuning of each IT site to all objects) that we had previously fit to real IT data. Therefore, the distribution of size tuning in the simulations are, by construction, matched to the distribution of size tuning observed in IT (Figure 3—figure supplement 1B and C). Because of this randomness in the size tuning kernel selected for each neural site and the variance introduced for the site’s responses to different image exemplars, the response profile of each simulated neuron is unique and thus the update direction during simulated unsupervised learning is not always in the same direction (Method: lines 1354-1358). To demonstrate this point more clearly, we’ve added another simulated IT site in Figure 3A, and added example IT recordings across size and objects in Figure 3—figure supplement 1A.

Modified text in methods (lines 1354-1358):

“Because of the initial randomness in the size tuning kernel selected for each neural site (Figure 3—figure supplement 1B and C) and the clutter variance introduced for the site’s responses to different image exemplars (Figure 3—figure supplement 1D), the response profile of each simulated neuron is unique and thus the update direction during simulated unsupervised learning is not always in the same direction across the population.”

Is the improvement of discrimination performance in the model independent of the specific size tuning template that is used? Or does it depend on whether the firing rate at the large size is higher or lower than at the medium size? An independent sampling of size tuning for the different neurons would have been the more natural choice. It would be comforting to see the results hold up.

As clarified before, we did independent sampling for size tuning curves. Therefore, the results presented here are based on random sampling of size tunings for different neurons, which is exactly what the reviewers suggested here.

However, since it caused confusion in the previous version, we have now added another simulated IT example in Figure 3 to reflect the random sampling and rephrased the method of size tuning creation and selection in the figure legend, result section and method section to emphasize this point.

In the figure legend: “Then, a size tuning kernel is randomly drawn from the a pool of size tuning curves (upper right box; kernels fit to real IT data) and multiplied by the object response distribution (outer product), resulting in a fully size-tolerant (i.e. separable) neural response matrix (64 objects x 3 sizes).”

In the result section: “To simulate the variance in object size, for each simulated site, we randomly chose one size tuning kernel from a pool of size tuning curves that we had obtained by fitting curves to real IT responses across changes in presented object size (n=168 recording sites; data from (Majaj et al., 2015)). This process is repeated independently for each simulated site.”

In the method section: “To generate the simulated IT tuning over changes in object size, for each simulated IT site, we multiplied (outer product) that 64x1 vector with a randomly chosen size tuning kernel (1x3 vector) that was randomly sampled from a batch of size tuning curves that we had obtained by fitting curves to real IT responses across changes in presented object size (n=168 recording sites; data from (Majaj et al., 2015)).”

Previously in the model schematic in Figure 3 there was only one simulated IT site and could have caused confusion, so we have added one more example simulated site to the figure to show the fact that each simulated IT site has a size tuning kernel randomly drawn from the pool of kernels fit to real IT data.

Please note that one of the reviewers actually went through the trouble to actually quickly code up a toy variant of the model. They reported that it appeared as if the learning rate in the model could be controlled by means of setting noise levels, and the consistent size tuning altered how much the model's classification performance improves for the non-swap exposure scenario. It was also noted that the improvement for non-swap and the deterioration for swap is actually even more pronounced without the consistency in size tuning, which maybe speaks in favor of at least the qualitative aspect of the model. The question thus remains whether it is possible to get both learning rates to match the data without the tuning consistency. This is an important point raised and we expect additional comments and controls.

We are very impressed and thankful that the reviewers’ took the time to simulate a variant of the overall model! However, we think it should be clear at this point that we didn’t use fixed size tuning and our model didn’t assume tuning consistency. The results presented here represent random draw from the size tuning kernels extracted from real data. We have also explained how we optimized the learning rate for our simulated IT population and what factors can influence its value (specifically the IT population statistics could influence the value of learning rate) at the beginning of this question. To further clarify, the learning rate value is specific to the IT population statistics and may not be a universal value. As also stated in the reply above, we do not view the plasticity rate that we report as a key claim of our study. We’ve now added clarification in the text (Methods: lines 1356-1360).

We are fully aware that the IT representation (simulated with the multi-dimensional Gaussian model), size tuning kernels (variance across size) and noise levels (variance caused by samplers or repeats) can influence the learning rate value. We modified result and method section to make this point clear.

Result lines 295-298:

“The simulated IT population responses were all constrained by recorded IT population statistics (Figure 3—figure supplement 1). These statistics define the initial simulated IT population response patterns and thus they ultimately influence the predicted unsupervised neural plasticity effects and the predicted behavioral consequences of those neural effects.”

Methods: lines 1356-1360

“However, it is important to note that the learning rate value is determined by the IT population statistics and the plasticity rule chosen here and thus should not be taken as a universal value. Changes in the statistics of the simulated IT population (i.e. covariance matrix, variance across sizes or clutter variance, etc) can influence the initial state of the IT population, and as a consequence influence both plasticity rate value and the predicted changes for each simulated IT neuron.”

[Editors' note: further revisions were suggested prior to acceptance, as described below.]

Essential Revisions:As requested by Reviewer #2, please:1) Add a brief discussion regarding differences in how objects/faces are encoded in the visual cortex and possible implications (or lack thereof) for the interpretation of results.

We interpret the reviewer’s comment as asking us to update the manuscript to discuss the possibility that the unsupervised plasticity effects of IT neurons are equivalent for all face-selective neurons, regardless of if those neurons are clustered within face patches (monkeys; within FFA or OFA in humans) or are scattered throughout IT. As such, we have added the following paragraph to the discussion (lines 735-755):

“Our study tested unsupervised learning effects for multiple types of object discriminations (Figure 6). […] More targeted future neurophysiology experiments could test these specific alternative hypotheses.”

2) Confirm that corrections for multiple comparisons were implemented.

When conducting multiple tests of the same null hypothesis and considering any one of those tests to reject that null hypothesis, this results in an increase in the likelihood of incorrectly rejecting the null hypothesis by pure chance. To set an appropriate null rejection level, a correction for multiple comparison (e.g. Bonferroni correction or FDR) is conducted, which corrects the α level for each test to account for the number of tests of the same null hypothesis.

In our testing of learning effects over exposure amount (Figure 2), we are not asking whether the learning effect for any exposure amount is different from 0, which would require multiple comparison correction for number of tests. Instead, each point is a single test of each null hypothesis: “There is no learning effect at exposure amount x.” Therefore, we do not believe the Bonferroni correction is applicable in this situation, and we leave it to the reader to decide on what inferences to draw from the data, confidence intervals and associated statistical tests. If the reviewer prefers, we could only test/show the maximum exposure point for panels A and B as that is the planned test of the simple null hypothesis: “Unsupervised exposure of this type for the maximum duration we could supply causes no change in discrimination performance."

In the statistical test for learning effect of different tasks (Figure 6), the dependent variable is the observed learning effect for tasks that differ in initial task difficulty. Here, we are asking whether the learning effect that is measured at 800 exposures for a given task is significantly different from 0 (the null hypothesis for all tasks). Thus, there is only one comparison for each task, therefore we believe that a multiple comparisons correction is not necessary here. If we were asking whether there is any learning effect observed for any one of the measured tasks, then a Bonferroni correction would be necessary, but that is not the question being asked here. Instead, we are simply showing the trend of the effect size for each tested task, with bootstrapped standard deviations of the mean, and leaving it to the reader to draw inferences about the relationship between initial task difficulty and learning effects size (the main goal of this figure). Indeed, if the reviewer prefers, we could drop the significance testing symbols from Figure 6 without loss of content.

[Editors' note: further revisions were suggested prior to acceptance, as described below.]

The manuscript has been improved but there are some remaining issues that need to be addressed, as outlined below:The argument that you are making regarding the absence of correction for multiple comparisons sounds reasonable, but it is open to debate. While we are sympathetic to your point about letting "the reader [to] decide on what inferences to draw from the data, confidence intervals and associated statistical test", we also feel that in the current manuscript form, the reader might not realize the absence of corrections. As a compromise, we suggest that you add a paragraph in the methods including your justification to make this clear.

We understand whether or not to apply correction for multiple comparisons in our case may be open to debate in the field. As suggested by the editor, we have added our justification in the Method Section to make it clear why we didn’t conduct correction for multiple comparison for our analysis. Please see the modified text below (lines 1300-1317):

“When conducting multiple tests of the same null hypothesis and considering any one of those tests to reject that null hypothesis, this results in an increase in the likelihood of incorrectly rejecting the null hypothesis by pure chance. […] Instead, we are simply showing the trend of the effect size for each tested task, with bootstrapped standard deviations of the mean, to demonstrate the relationship between initial task difficulty and the learning effect size. ”